# Prediction of monthly Arctic sea ice concentrations using satellite and reanalysis data based on convolutional neural networks

Young Jun Kim[1], Hyun-cheol Kim[2], Daehyeon Han[1], Sanggyun Lee[3], and Jungho Im[1]

[1]School of Urban and Environmental Engineering, Ulsan National Institute of Science and Technology, Ulsan, South Korea
[2]Unit of Arctic Sea-Ice Prediction, Korea Polar Research Institute, Incheon, Republic of Korea
[3]Centre for Polar Observation and Modelling, University College London, London, UK

*Correspondence to*: Jungho Im (ersgis@unist.ac.kr)

**Abstract.** Changes in Arctic sea ice affect atmospheric circulation, ocean current, and polar ecosystems. There have been
unprecedented decreases in the amount of Arctic sea ice, due to global warming and its various adjoint cases. In this study, a novel one-month sea ice concentration (SIC) prediction model is proposed, with eight predictors using a deep learning approach, Convolutional Neural Networks (CNN). This monthly SIC prediction model based on CNN is shown to perform better predictions (mean absolute error (MAE) of 2.28%, anomaly correlation coefficient (ACC) of 0.98, root mean square error (RMSE) of 5.76%, normalized RMSE (nRMSE) of 16.15%, and NSE of 0.97) than a random forest (RF)-based model
(MAE of 2.45%, ACC of 0.98, RMSE of 6.61%, nRMSE of 18.64%, and NSE of 0.96) and the persistence model based on the monthly trend (MAE of 4.31%, ACC of 0.95, RMSE of 10.54%, nRMSE of 29.17%, and NSE of 0.89) through hindcast validations. The spatiotemporal analysis also confirmed the superiority of the CNN model. The CNN model showed good SIC prediction results in extreme cases that recorded unforeseen sea ice plummets in 2007 and 2012 with less than 5.0% RMSEs. This study also examined the importance of the input variables through a sensitivity analysis. In both the CNN and RF models,
the variables of past SIC were identified as the most sensitive factor in predicting SIC. For both models, the SIC-related variables generally contributed more to predict SICs over ice-covered areas, while other meteorological and oceanographic variables were more sensitive to the prediction of SICs in marginal ice zones. The proposed one-month SIC prediction model provides valuable information which can be used in various applications, such as Arctic shipping route planning, management of fishery industry, and long-term sea ice forecasting and dynamics.

## 1 Introduction

Sea ice refers to the frozen seawater that covers approximately 15% of the oceans in the world (National Snow and Ice Data Center, 2018). Sea ice reflects more solar radiation than the water's surface, which makes the polar regions relatively cool. Sea ice shrinks in summer due to the warmer climate and expands in the winter season. Many studies on Arctic sea ice monitoring and dynamics have been conducted because it plays a significant role in the energy and water balance of global
climate systems (Ledley, 1988; Guemas et al., 2014). In particular, the change in sea ice is an important indicator that shows

the degree of on-going climate change (Johannessen et al., 2004). Global warming causes a decrease in sea ice that worsens the arctic amplification, which in turn accelerates global warming itself (Cohen et al., 2014; Francis and Vavrus, 2015). In addition, sea ice affects various oceanic characteristics and societal issues, such as ocean current circulation, by changing salinity and temperature gradation (Timmermann et al., 2009); polar ecosystems, by affecting key parts of the Arctic food web like sea-ice algae (Doney et al., 2011); and economic industries e.g., Arctic shipping routes (Melia et al., 2016).

Arctic sea ice has been rapidly declining, which impacts not only the Arctic climate but also possibly the mid-latitudes (Yu et al., 2017). Numerous studies have shown significant interactions between the ocean and climate characteristics, such as sea surface temperature, solar radiation, surface temperature, and the changes in sea ice (Guemas et al., 2014). Therefore, the prediction of long and short-term sea ice change is an important issue in projecting climate change (Yuan et al., 2016). Various approaches, including numerical modeling and statistical analysis, have been proposed to develop models to predict sea ice characteristics (Guemas et al., 2014; Chi and Kim, 2017). Many of the studies have adopted statistical models using *in situ* observations or reanalysis data based on the relationship between sea ice and ocean/climate parameters (Comeau et al., 2019). The long-range forecasting models of sea ice severity index and concentration (monthly to seasonal) using multiple linear regression were developed by Drobot (2003) and Drobot et al. (2006), respectively. Lindsay et al. (2008) examined the short and long-term sea ice extent (SIE) prediction using a multiple linear regression model with historical information regarding the ocean and ice data. Wang et al. (2016) developed a vector autoregressive (VAR) model to predict the intraseasonal variability of SIC in the summer season (May – September). The suggested VAR model considering only the historical sea ice data without any atmospheric and oceanic information showed a root mean square error (RMSE) ~ 17% for 30-days' prediction. However, the literature has reported that sea ice prediction is a very challenging task under the changing Arctic climate system (Holland et al., 2010; Stroeve et al, 2014). A short-term forecast of SIC has been also examined using statistical approaches. Wang et al. (2019) evaluated the sub-seasonal predictability of Arctic SIC using multi-variables of sea ice, the atmosphere, and the ocean based on statistical approaches—the VAR and vector Markov models. The VAR model showed quite good predictability in the short-term with RMSE of 10%, but still resulted in high RMSEs (~20%) for longer than 4 weeks over pan-Arctic during the summer season (from June to August). Meanwhile, the Data-Adaptive Harmonic (DAH) technique, which examines a data-driven feature using cross-correlations, was demonstrated to predict Arctic SIE (Kondrashov et al., 2018). The DAH model showed a promising predictability of SIE in September, resulting in the absolute error of about 0.3 million $km^2$ in 2014-2016. Chi and Kim (2017) suggested a deep learning-based model using Long and Short-Term Memory (LSTM) in comparison to a traditional statistical model. Their model showed good performance in the one-month prediction of sea ice concentration (SIC), with less than 9% average monthly prediction errors. However, it had low predictability during the melting season (RMSE of 11.09% from July to September). Kim et al. (2018) proposed a near-future SIC prediction model (10-20 years) using deep neural networks together with the Bayesian model averaging ensemble, resulting in RMSE of 19.4% on the annual average. This study suggests that deep learning techniques are good to connect variables under non-linear relationships, such as SIC and climate variables. However, this study also showed low prediction accuracy during the melting season

(nRMSE of 102.25% from June to September). Wang et al. (2017) used convolutional neural networks (CNN) to estimate SIC in the Gulf of Saint Lawrence from synthetic aperture radar (SAR) imagery. Their study compared their CNN model to a multilayer perceptron (MLP) model, showing the superiority of the CNN model in SIC estimation with an RMSE of about 22%.

However, different from the classic statistical models, the previous studies using deep learning techniques have focused on the long-term prediction of SIC (i.e., over one-year prediction). The short-term forecasting of sea ice conditions is also important for maritime industries and decision making on-field logistics (Schweiger and Zhang, 2015). In addition, there is room to further improve the accuracy of short-term SIC prediction models with more advanced techniques and data. SIC describes the fraction of a specified area (typically a grid cell) covered by sea ice and it has been widely used as a simple and intuitive proxy to identify the characteristics of sea ice. Thus, this study aimed to predict the changes in Arctic sea ice characteristics using SIC.

This study proposes a novel deep learning-based method to predict SIC based on the predictors of spatial patterns, considering the operational forecast of sea ice characteristics. The objectives of this study were to (1) develop a novel monthly SIC prediction model using a deep learning approach, CNN; (2) examine the prediction performance of the proposed model through comparison with a random forest-based SIC prediction model; and (3) conduct a sensitivity analysis of predictors that affect SIC predictions.

## 2. Data

Three types of datasets were used in this study, which represent sea ice concentrations, oceanographic, and meteorological characteristics in the Arctic. This study focuses on the prediction accuracy of the proposed models as well as the sensitivity of each predictor on monthly SIC prediction. The spatial domain of this study is a region of the Arctic Ocean (180°W – 180°E / 40°N – 90°N), and the temporal coverage is the 30 years between 1988 and 2017.

The first dataset is the daily sea ice concentration observation data, obtained from the National Snow and Ice Data Center (NSIDC), which is derived from the Nimbus 7 Scanning Multichannel Microwave Radiometer (SMMR) and the Defense Meteorological Satellite Program (DMSP) Special Sensor Microwave Imager (SSM/I and SSMIS). The second dataset is the daily sea surface temperature data, obtained from National Oceanic and Atmospheric Administration (NOAA) Optimal Interpolation Sea Surface Temperature (OISST) version 2, which is constructed from Advanced Very High-Resolution Radiometer (AVHRR) observation data with 0.25° resolution from 1988 to 2017. The third dataset is the monthly European Centre for Medium-Range Weather Forecasts (ECMWF) Re-Analysis Interim (ERA-Interim) data, which is used in order to construct predictors for one-month SIC prediction, including the surface air temperature, albedo, and v-wind vector in 0.125°.

In this study, a total of eight predictors were selected and used to predict SIC next month (Table 1) based on the literature and a preliminary statistical analysis of potential predictors through a feature selection process using random forest (Strobl et al., 2007). We selected the eight predictors by comparing the mean decrease accuracy (MDA) changes based on twelve monthly prediction RF models from 1988 to 2017. The MDA has been widely used as feature selection criteria by measuring the accuracy changes by randomly permuting input variables (Archer and Kimes, 2008). It should be noted that fewer predictors than the selected eight ones did not produce better results. The predictors are: SIC one-year before (sic_1y), SIC one-month before (sic_1m), SIC anomaly one-year before (ano_1y), SIC anomaly one-month before (ano_1m), sea surface temperature (SST), 2-meter air temperature (T2m), forecast albedo (FAL), and the amount of v-wind (v-wind).

**Table 1.** The specifications of the eight predictors used to predict short-term SIC in the study.

| Variable | Source | Unit | Temporal resolution | Spatial resolution | Normalization |
|---|---|---|---|---|---|
| SIC one-year before (sic_1y) | NSIDC | % | Daily | 25km | 0 - 1 |
| SIC one-month before (sic_1m) | NSIDC | % | Daily | 25km | 0 - 1 |
| SIC anomaly one-year before (ano_1y) | NSIDC | % | Daily | 25km | -1 - 1 |
| SIC anomaly one-month before (ano_1m) | NSIDC | % | Daily | 25km | -1 - 1 |
| Sea surface temperature one-month before (SST) | NOAA OISST ver.2 | K | Daily | 0.25° | 0 - 1 |
| 2-meter air temperature one-month before (T2m) | ECMWF ERA Interim | K | Monthly | 0.125° | 0 - 1 |
| forecast albedo one-month before (FAL) | ECMWF ERA Interim | % | Monthly | 0.125° | 0 - 1 |
| the amount of v-wind one-month before (v-wind) | ECMWF ERA Interim | m/s | Monthly | 0.125° | 0 - 1 |

In order to have the same spatial and temporal scales, the daily data, including SIC and SST, were transformed into monthly-means and onto a polar stereographic projection with 25km grids. The predictors were normalized into 0 to 1 or -1 to 1 (for ano_1y and ano_1m). Since sea ice decline has accelerated in recent years, especially in the summer season (Stroeve et al, 2008; Schweiger et al., 2008; Chi and Kim, 2017), we computed the SIC anomaly variables only for a more recent time period (2001-2017) rather than the entire study period (1988-2017). This was done in order to focus on the trends in recent sea ice changes. Since the anomalies were calculated from the recent years (2001-2017), there is no significant multicollinearity issue that could cause overfitting (Pearson's correlation coefficient between mean SICs and anomalies ($\rho$) = -0.39, $p<0.01$). The v-wind indicates the relative amount of wind towards the North Pole: the larger the v-wind, the more it blows from South to North. The v-wind data were derived using an 11-by-11 moving window based on a mean function from the raw 10-meter-height v-wind vector data. Regarding the moving window, this study set the analysis unit as an 11-by-11 window (neighboring 5 pixels; about 125 km) in order to consider the synoptic-scaled climate and ocean circulation in the polar region (Crane, 1978; Emery et al., 1997).

The eight predictors selected in this study through random forest-based feature selection have theoretical backgrounds that are related to the characteristics of SIC. First, SIC itself can affect the SIC in the future because it has a clear inter-annual trend through the melting and freezing seasons (Deser and Teng, 2008; Chi and Kim, 2017). It is a useful characteristic when conducting a time-series analysis, and thus, two SIC time-series climatology predictors (SIC one-year before and SIC one-month before) were used in this study. Although there is no clear physical explanation of why the interannual variations would contribute to the forecasting skill, it clearly worked well in long-term SIC forecasting in previous studies (Wang et al., 2016; Chi and Kim, 2017). Further, we used two supplementary predictors that indicate the anomalies of SIC one-year before and SIC one-month before, in order to consider anomalous sea ice conditions in the models. The anomaly data could give information about SST anomaly along the sea ice edge in terms of the re-emergence mechanism from the melting to the freezing seasons (Guemas et al., 2014). Second, changes in SST and SIC have a significant relationship with each other, with regards to the heat budget (Rayner et al., 2003; Screen and et al., 2013; Prasad et al., 2018). The re-emergence of sea ice anomalies is also partially explained by the persistence of SST anomalies (Guemas et al., 2014). Air temperature and albedo are related to the amount of solar radiation enabling the prediction of SIC changes. The solar radiation heats the surface of the ocean as well as the sea ice. This causes a rise in the SST while also reducing albedo on the sea ice by melting the surface snow or thinning the sea ice (Screen and Simmonds, 2010; Mahajan et al., 2011). Moreover, the surface snow melting produces melt ponds, wet sea-ice surfaces, and wet snow cover which accelerate sea ice melting (Kern et al., 2016). Warm winds from lower latitudes toward the Arctic can also reduce sea ice (Kang et al., 2014) and local wind forces affect sea ice motion and formation (Shimada et al., 2006). The wind vector also can cause short or long-range sea ice drifts (Guemas et al., 2014), which may influence SIC variation.

## 3. Methods

### 3.1 Prediction models: Convolutional Neural Networks (CNN), Random Forest (RF), and anomaly persistence model

This study proposes a SIC prediction model using a Convolutional Neural Network (CNN) deep learning approach. CNN is a kind of artificial neural network (ANN) model first suggested by LeCun et al. (1998) and has since been further developed with various structures and algorithms (Deng et al., 2013). Many studies have adopted CNN approaches to complete image recognition or classification tasks (Kim et al., 2018a; Ren et al., 2015; Yoo et al., 2019; Zhang et al., 2019b). CNN learns the features of images and takes them into account as key information, in order to extract outputs (Kim et al., 2018b; Wiley et al., 2019). Convolutional networks share their weights and connect neighboring layers using convolution layers like neurons (Yu et al., 2017). The convolutional structure is a unique feature of CNN models that often shows higher performance than other types of ANN in image recognition studies (Krizhevsky et al., 2012; Lee et al., 2009; Zhao et al., 2020). The basic CNN structure consists of a bundle of convolutional layers, a number of pooling layers, and a fully connected layer. The convolutional process is to generate feature maps from gridded input data with kernel and activation functions. A CNN model

extracts the best feature map from an input image through an iterative training process including backpropagation learning and optimization algorithm.

In CNN approaches, when 3-dimensional data (i.e., width, height, and depth (or channel)) are entered, several moving kernels pass through the data for each channel and transform them into feature maps using dot-product calculation. Through a number of convolutional processes, the model uses the fully connected layer to generate the final answer. The series of convolutional processes involved in this process requires significant computation loads. To prevent heavy computation, both the stride (i.e., how to shift a moving kernel) and the pooling (i.e., how to conduct downsampling) techniques are widely used, which make the size of the input data in the following convolutional process reduced. To avoid too much data reduction, many studies have adopted a padding technique, which covers input data with extra dummy values (Wang et al., 2016). The feature map achieved through the convolutional process is a convolved map that contains a higher level of features of an image (Chen et al., 2015). In general, a CNN model contains larger learning capacity and provides more robustness against noise than normal MLP models because of the more trainable parameters as well as the structure of deeper networks (Wang et al., 2017).

In order to conduct a quantitative comparison of the prediction performance of the proposed CNN model, this study used random forest (RF), which is an ensemble-based machine learning technique (Latifi et al, 2018; Yoo et al., 2018). The RF model was used to solve image-based classification problems such as building extraction, land-cover classification, and crop classification (Liu et al., 2018; Guo and Du, 2017; Forkuor et al., 2018; Sonobe et al., 2017). RF extracts features using classifiers of each variable (Zhang et al., 2019a). The user can deal with two main parameters: the number of decision trees and the number of split variables at the nodes (Fagua et al., 2019). In this study, we used 50 trees and 11 random variables to be used in the decision split because random selection using one-third of variables in each split has been used widely in solving regression problems (Chu et al., 2014; Mutanga et al., 2012; Mutowo et al., 2019). Compared to the CNN approach, RF has a relatively low learning capacity from the perspective of the parametric size.

Finally, an anomaly persistence forecast model was also examined for predicting the monthly Arctic SIC. The anomaly persistence model is a useful reference for forecast skill for time-series data (Wang et al., 2016). Since sea ice shows a clear climatological pattern (Parkinson and Cavalieri, 2002; Deser and Teng, 2008; Chi and Kim, 2017), this study used the persistence forecast model along with the RF regression model as baseline models to figure out the performance of the CNN model for SIC prediction.

## 3.2 Research Flow

This study examined three models in order to predict SIC using the persistence and RF-based (baselines), and CNN-based approaches (Fig. 1). We designed twelve individual models (i.e., monthly models) to predict SIC for each month. A hindcast validation approach was used to evaluate each model's performance. Each monthly model was trained using the past data

staring from 1988. For instance, 12-years' data (1988-1999) and 29-years' data (1988-2016) were trained to predict SICs in 2000 and 2017, and 2000 and 2017 SIC data were used as validation data, respectively. Eight input data during the past 30 years that consist of 304 * 448 sized grids were used as training data in the RF and CNN models. In the case of the RF model, an additional 24 input parameters, along with the eight predictors, were considered. They are the mean, minimum, and maximum values of each predictor calculated using the 11-by-11 window. These additional variables for RF are to fill the conceptual gaps between the two approaches by considering the spatial patterns of predictors such as features in the CNN model. Since most SIC samples were biased to zero values because of the numerous pixels in the open sea, the training samples were balanced out considering the SIC values (0 – 100%) using a monthly maximum SIE mask, which shows the widest SIE during the entire study period (1988-2017) for each month. As a result, in the case of 2017, about 600,000 samples on average (i.e., from about 400,000 samples in Sep. to about 850,000 samples in Mar.) were trained for both monthly models (i.e., RF and CNN). However, the unbalance sampling problem still remained because the lower SIC (less than 40%) samples were relatively small (about 20% of the entire training samples). In the case of the anomaly persistence forecast model, the monthly SIC anomaly of each pixel persisted and the observed trend was calculated for that month ahead. For example, SICs in Jan. 2000 were predicted by summing one-month persisted anomaly and one-month ahead SIC from a linear trend of SICs from Jan. 1988 to Dec. 1999 by each grid.

As described in Fig. 1, the CNN model consists of three convolutional layers and one fully connected layer. Wang et al. (2017) used CNNs to estimate SIC from SAR data and showed that the use of three convolutional layers performed better than one or two layers. In this study, the root mean square propagation (RMSProp) optimizer with a learning rate of 0.001 and the relu activation function were used in the model. The RMSProp optimizer has a similar process to a gradient descent algorithm which divides the gradients by a learning rate (Tieleman and Hinton, 2012). Fifty (50) epochs with batch size as 1,024 were used in the proposed CNN model. The best model showing the highest validation accuracy during the training process was selected and used for further analysis. The CNN model was implemented using the Tensorflow Keras open-source library, while the persistence and RF models were implemented using the interp1 and TreeBagger functions in the MATLAB r2018a, respectively.

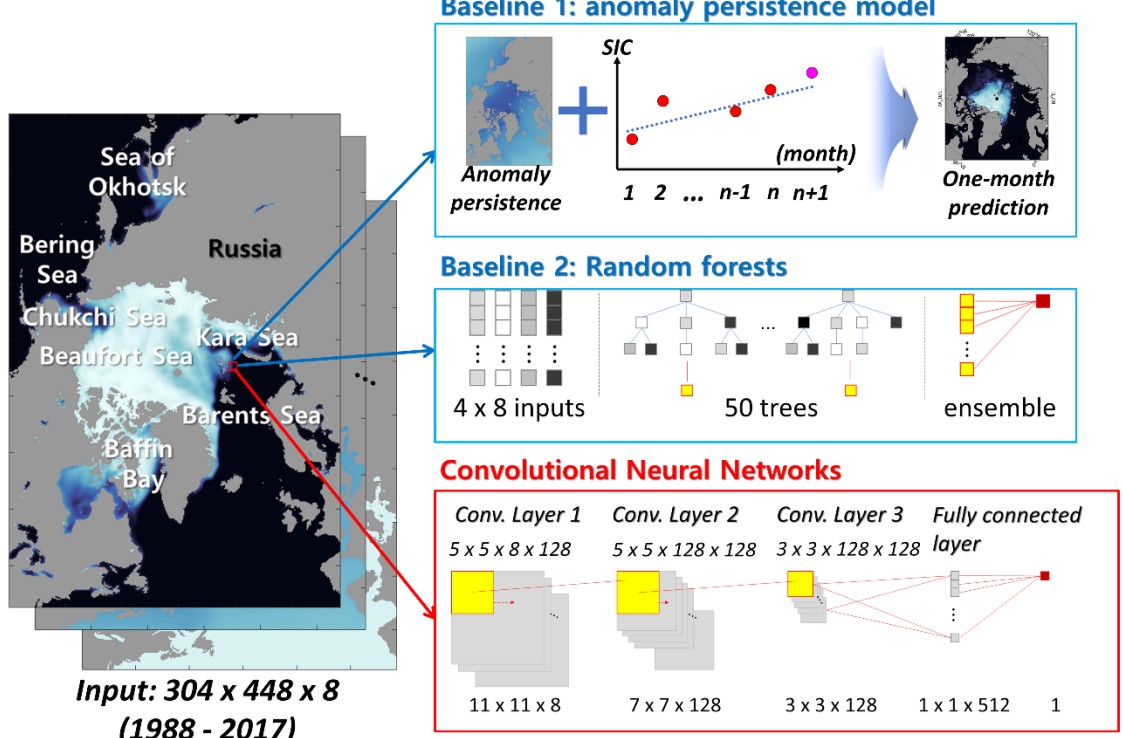

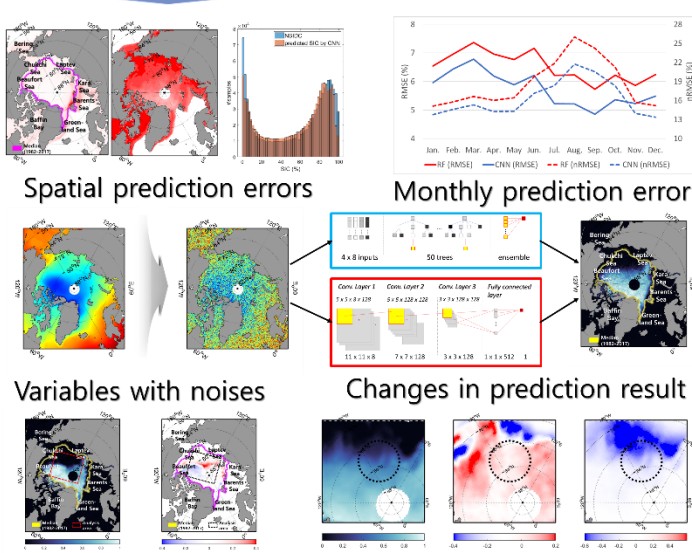

**Figure 1.** Study area and research flow.

This study firstly evaluated the model performance by quantitatively comparing the prediction results of the three models based on five accuracy metrics: mean absolute error (MAE, Eq. (1)), anomaly correlation coefficient (ACC, Eq. (2)), root mean square error (RMSE, Eq. (3)), normalized root mean square error (nRMSE, Eq. (4)), and Nash-Sutcliffe efficiency (NSE, Eq. (5)). In the melting season, many pixels contain relatively low SIC values compared to the freezing season. By dividing the RMSE by the standard deviation of actual SICs, the nRMSE can represent the prediction accuracy considering the range of SIC values (Kim et al., 2018). The ACC is a measure of skill score to evaluate the quality of the forecast model (Wang et al., 2016) and has a value between -1 (inversely correlated) and 1 (positively correlated). The NSE is a widely-used measure of prediction accuracy (Moriasi et al., 2007). It can provide comprehensive information regarding data by comparing the relative variance of prediction errors and the variance of the observation data (Nash and Sutcliffe, 1970; Moriasi et al., 2007). The NSE has a range from $-\infty$ to 1.0. A model is more accurate when the NSE value closer to 1, but unacceptable when the value is negative (Moriasi et al., 2007). Every error matrix was computed with respect to space and time. The errors were spatially averaged after masking, and then temporally averaged.

$$MAE = mean(|predicted\ SIC - actual\ SIC|) \tag{1}$$

$$ACC = \frac{mean(\sum(predicted\ SIC - \overline{predicted\ SIC})(actual\ SIC - \overline{actual\ SIC}))}{\sqrt{mean(\sum(predicted\ SIC - \overline{predicted\ SIC})^2)}\sqrt{mean(\sum(actual\ SIC - \overline{actual\ SIC})^2)}}, \bar{x}: mean \tag{2}$$

$$RMSE = \sqrt{mean[(predicted\ SIC - actual\ SIC)^2]} \tag{3}$$

$$nRMSE = \frac{\sqrt{mean[(predicted\ SIC - actual\ SIC)^2]}}{std(actual\ SIC)} \tag{4}$$

$$NSE = 1 - \frac{\sum(actual\ SIC - predicted\ SIC)^2}{\sum(actual\ SIC - mean(actual\ SIC))^2} \tag{5}$$

With respect to prediction accuracy analysis, a specific mask that covers only pixels that have shown sea ice more than once in the past 10 years was used to prevent an inflation of overall accuracy that may have happened due to the effect of pixels on open seas in the melting season (Chi and Kim, 2017; Kim et al. 2018). For example, to calculate the prediction accuracy of predicted SIC in January 2017, the mask covered only pixels that have shown sea ice in Januarys from 2007 to 2016. To examine prediction performance in the marginal sea ice zone, the models were compared in two cases: all range of SICs (0-100%) and low SICs (0-40%).

In addition, the study examined the spatial distribution maps showing the annual MAE and ACC of three models from 2000 to 2017. The spatial relationship between SIC anomalies and prediction errors was also explored. Since the actual anomalies, as well as actual prediction errors (predicted SICs – actual SICs), tended to cancel each other out by averaging negative and positive values, we used absolute anomaly and error values. Since the actual anomalies, as well as actual prediction errors (predicted SICs – actual SICs), tended to cancel each other out by averaging negative and positive values, we used absolute anomaly and error values. In order to examine temporal forecast skill, this study compared the ACC between the monthly time

series of reference and predicted SICs at each grid (Wang et al., 2016). The distribution of predicted SICs by both models was also compared for the melting season (Jun. – Sep.). The Sea Ice Outlook (SIO) open community has investigated the pan-Arctic sea ice especially in the September SIE since 2008 (Stroeve et al., 2014; Chi and Kim, 2017). They have shared the predicted September SIE from June, July, and August based on a heuristic, statistical, dynamical, and mixed approaches. Chi and Kim (2017) have pointed out the difficulties of sea ice prediction because the prediction errors have increased since 2012. To figure out September minimum SIE which is the main focus of the SIO community (Stroeve et al., 2014), we compared the predicted SIEs based on the three models evaluated in this study, together with the other 37 SIO contributions for the September SIE predictions reported in August 2017. In the present study, the SIE was identified as an area of SIC > 15% (Chi and Kim, 2017). Further, the averaged monthly trends of prediction accuracy using RMSE and nRMSE together were examined with the trends of annual mean nRMSE by dividing the data into melting (Jun. – Sep.) and freezing (Dec. – Mar.) seasons.

In this research, we compared and examined prediction results focusing on two extreme cases of SIC: September 2007 and 2012. There was unexpectedly large Arctic sea ice shrinkage in the summer 2007 and 2012 because of the large-scale changes in climate conditions and August cyclones, respectively (Devasthale et al., 2013). Therefore, for detailed analysis, visual interpretation comparing the spatial patterns of prediction errors and input variables was conducted by focusing on the regions showing high prediction errors in Sep. 2007 and Sep. 2012.

Finally, we examined the variable sensitivity for each model. Rodner et al. (2016) evaluated the variable sensitivity of built-in CNN architectures in three ways: adding random Gaussian noises, taking geometric perturbations, and setting random impulse noises (i.e., set the pixel values to zero) to input images. In this research, the analysis of variable sensitivity was conducted using their first and third methods. To examine the influence of variables on prediction accuracy, we added random Gaussian noises with zero-mean and 0.1 standard deviations, then compared any changes of RMSE for each variable (Eq. (6)). In addition, to examine the spatial effects on the predictions, the prediction results were compared by setting zero values for two groups of variables: sea ice related variables (sic_1y, ano_1y, sic_1m, and ano_1m) and other environmental variables (SST, T2m, FAL, and v-wind).

$$Sensitivity\ (Var_x) = \frac{Changed\ RMSE\ with\ variable\ x\ containing\ noises}{Original\ RMSE} \tag{6}$$

## 4. Results and Discussion

### 4.1 Monthly prediction of SIC

Table 2 shows the average prediction accuracies of the models from 2000 to 2017. The CNN model showed higher performance than the persistence as well as RF models in all accuracy metrics. When it comes to considering all range of SICs (0-100%), the persistence model resulted in the lowest prediction performance (MAE of 4.31%, ACC of 0.95, RMSE of 10.54%, nRMSE

of 29.17%, and NSE of 0.89). While the RF and CNN models resulted in good prediction accuracy with a small difference in MAE, ACC, and RMSE (CNN: MAE of 2.28%, ACC of 0.98, RMSE of 5.76%, and NSE of 0.97; RF: MAE of 2.45%, ACC of 0.98, RMSE of 6.61%, and NSE of 0.96), the CNN model showed better results than the RF model for nRMSE (16.15% and 18.64%, respectively). These results imply that the error distribution of the CNN model was more stable than the

persistence model as well as RF. For the low SICs (0-40%), the MAE increased but it was due to the lower SIC values. The RMSE and nRMSE of the persistence model have decreased, but the others increased (persistence: 8.94% of RMSE and nRMSE of 24.62%; RF: RMSE of 7.23% and nRMSE of 19.87%; and CNN: RMSE of 6.18% and nRMSE of 16.87%). It implies that the RF and CNN models might be relatively weak to predict SICs in the marginal sea ice zone when compared to the central zone. The ACC and NSE values decreased for all models for low SICs (persistence: ACC of from 0.95 to 0.54 and

NSE of from 0.89 to 0.81; RF: ACC of from 0.98 to 0.96 and NSE of from 0.96 to 0.90; and CNN: ACC of from 0.98 to 0.96 and NSE from 0.97 to 0.93). Especially, the persistence model shows a larger decrease than the other models. Nonetheless, the CNN model produced consistently higher performance than the other models for both cases.

**Table 2.** Average prediction accuracies among three models on every SIC (0-100%) and low SICs (0-40%) during 2000-2017 (mean absolute error, anomaly correlation coefficient, root mean square errors, normalized root mean square errors, and Nash-Sutcliffe efficiency).

|  |  | MAE | ACC | RMSE | nRMSE | NSE |
|---|---|---|---|---|---|---|
| All range of SICs (0-100%) | Persistence | 4.31% | 0.95 | 10.54% | 29.17% | 0.89 |
|  | RF | 2.45% | 0.98 | 6.61% | 18.64% | 0.96 |
|  | CNN | 2.28% | 0.98 | 5.76% | 16.15% | 0.97 |
| Low SICs (0-40%) | Persistence | 2.94% | 0.54 | 8.94% | 24.62% | 0.81 |
|  | RF | 2.38% | 0.96 | 7.23% | 19.87% | 0.90 |
|  | CNN | 2.13% | 0.96 | 6.18% | 16.87% | 0.93 |

The spatial distribution of the annual MAE of three models from 2000 to 2017 is shown in Fig. 2. From visual inspection, it appeared that the prediction errors were dominant in the marginal areas (i.e., the boundaries between the sea ice and open seas). Since the marginal sea ice, particularly thin ice, is susceptible to change (Stroeve et al., 2008; Chevallier et al., 2013; Zhang et al., 2013), the prediction accuracy may have decreased. Weak predictability on the marginal sea ice zone might be due to a relatively small training sample size over the area. In the melting season, relatively higher prediction errors appeared not only

in the marginal area, but also even ice-covered areas near the Arctic center (Fig. 2f-h). On the other hand, in the freezing season, the prediction errors were shown mainly in the marginal area (Fig., 2j-l). Further, relatively higher prediction errors appeared around the Kara Sea and the Barents Sea (Fig. 2a, e, and i). The region from the Kara Sea to the Barents Sea shows consistent sea ice retreats because of inflows of warm and salty ocean water from the Atlantic Ocean into the Barents-Kara Sea (Schauer et al., 2002; Årthun et al., 2012; Kim et al., 2018) and cumulative positive solar radiation in the summer season

(Stroeve et al., 2012). Using a visual comparison, it can be seen that the degree of errors is higher in RF than CNN (Fig. 2).

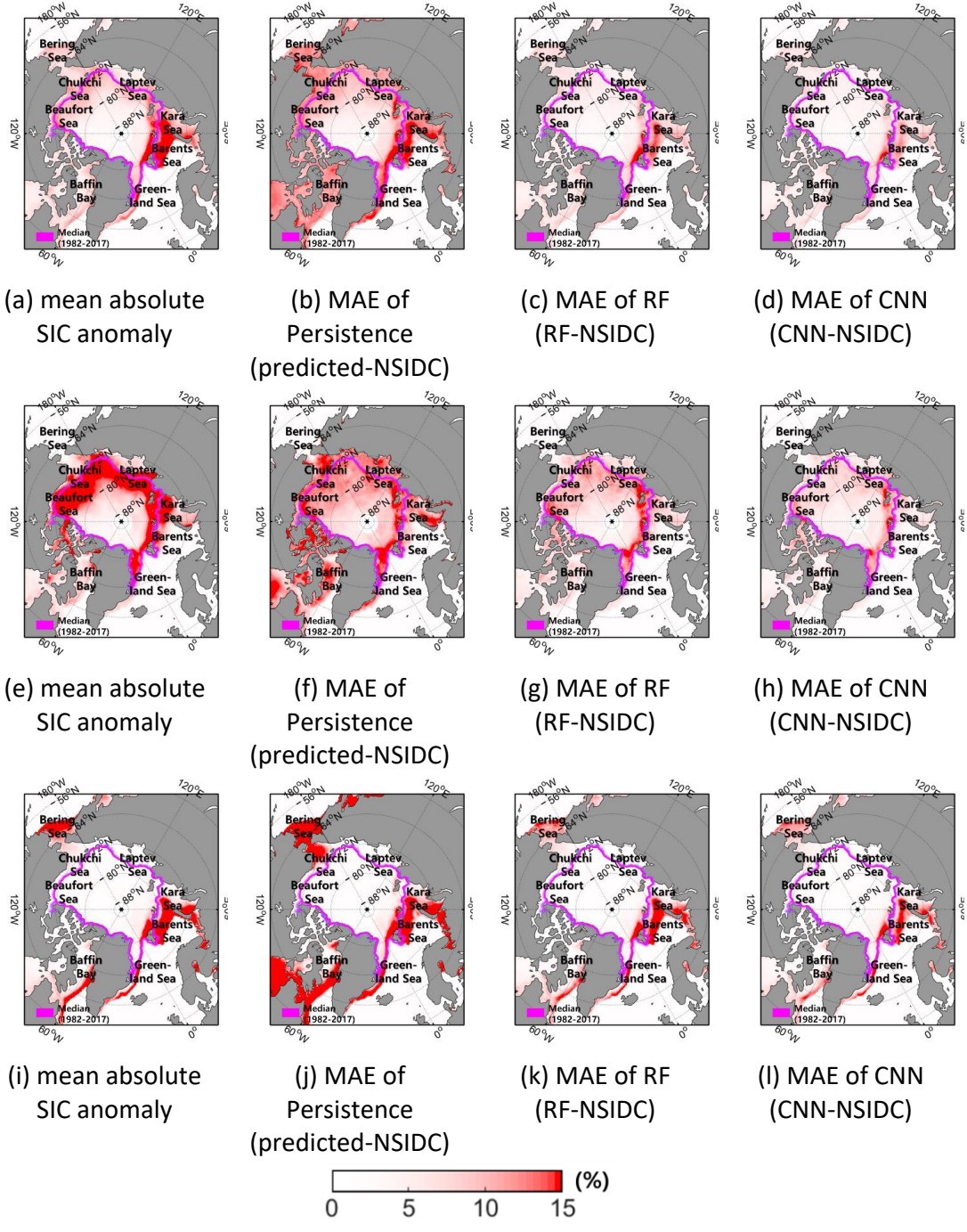

**Figure 2.** The mean absolute SIC anomaly (a) and mean absolute errors between predicted SICs and the actual SICs by the persistence (b), RF (c) and CNN (d) during 2000-2017. As in (a) - (d), but for the melting (Jun. – Sep.) and freezing (Dec. – Mar.) season, (e) - (f) and (i) - (l), respectively.

The spatial distribution of the temporal ACCs of three models from 2000 to 2017 is shown in Fig. 3. First of all, every prediction model showed quite good skill scores with high positive correlation (near 1.0, Fig. 3a-c). Interestingly, the ACCs were higher in the marginal area where showed relatively high prediction errors. Even though the models were weak to predict SIC changes in the marginal sea ice zone, but they caught decreasing trends of SICs relatively well. On the other hand, the region near the Arctic center showed relatively low ACCs. In contrast to the marginal sea ice zone, the Arctic center region is relatively stable

to the changes (Stroeve et al., 2008; Chevallier et al., 2013). Since SICs in the center is almost saturated (100% of SIC) and very stable, it might cause lower ACC values even there were relatively small prediction errors. In case of the melting season (Jun. – Sep., Fig. 3d-f), the degree of ACCs decreased when compared to the annual-mean (Fig. 3a-c), but they also showed the decreasing trends well in accordance with global warming. Unlike the melting season, the freezing season (Dec. – Mar.) showed relatively lower ACCs in the marginal and Arctic center regions (Fig. 3g-i). The persistence model did not catch the

decreasing trend and showed a negative correlation in the Laptev Sea (Fig. 3g). Further, the ACCs were quite low in the Arctic center region. As mentioned above, the stable and saturated sea ice resulted in lower skill scores in terms of ACC. From visual inspection, the CNN model showed better prediction with a stable skill score than the other models.

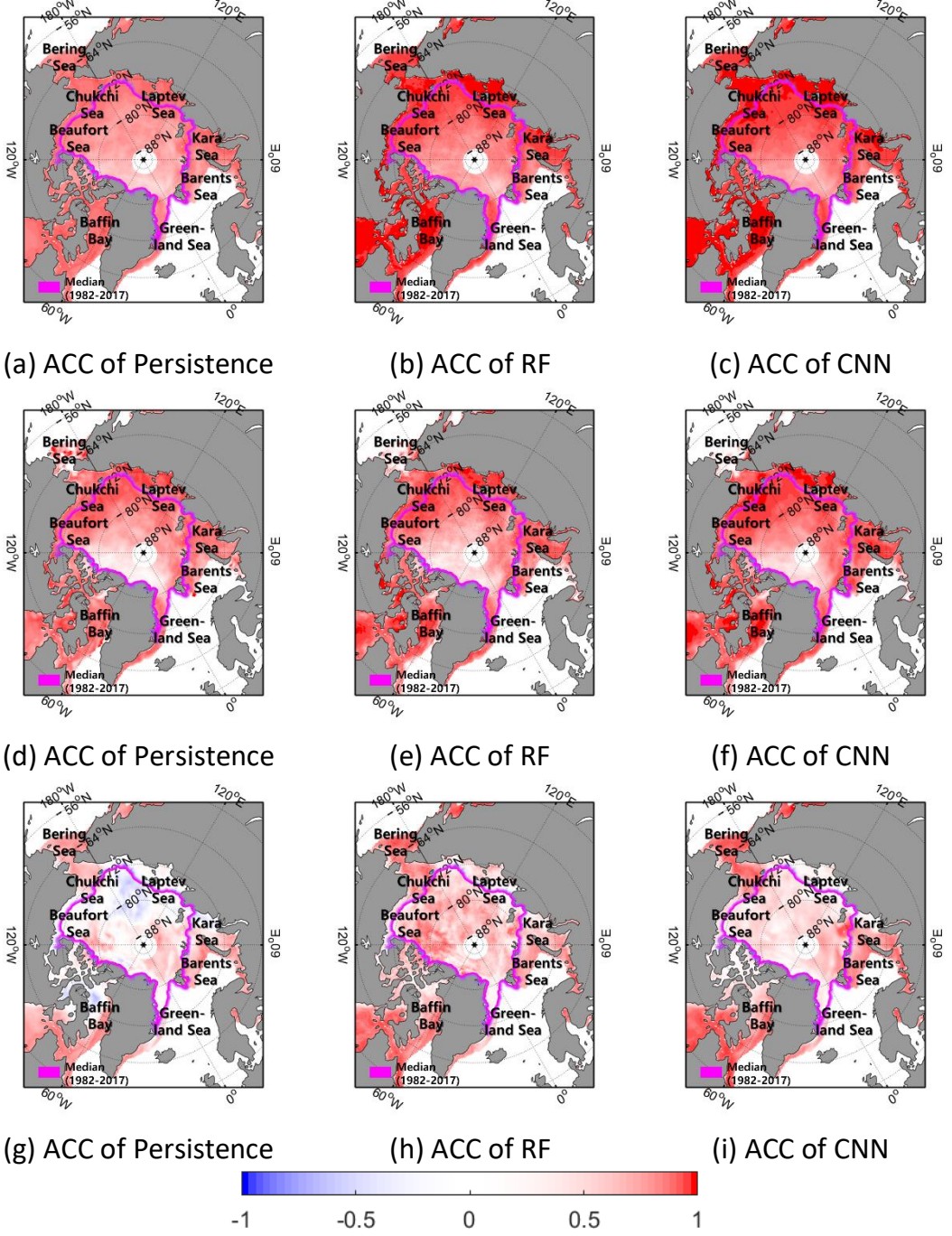

**Figure 3.** The temporal ACC of the persistence (a), RF (b) and CNN (c) during 2000-2017. As in (a) - (c), but for the melting (Jun. – Sep.) and freezing (Dec. – Mar.) seasons, (d) - (f) and (g) - (i), respectively.

Figure 4 shows the histograms of NSIDC SICs and the predicted SICs by three models in the melting season (Jun. – Sep.) during 2000-2017. The persistence forecasting model shows poor predictability for all ranges of SICs (Fig. 4a). In addition,
the model tended to over-estimate for higher SICs in the melting season. The model did not catch well the decreasing trends of sea ice due to global warming. On the other hand, the RF and CNN models showed relatively weak predictability for boundary SIC values (i.e., less than 10% and over 90% SICs). In particular, the RF model showed a weakness to predict SICs near zero (0%) and 100%. By focusing on the RF and CNN models, the mean and standard deviation values of prediction errors (predicted SIC - NSIDC) were examined for lower as well as higher SICs. In the case of lower SICs (less than 5%), both
models showed over-estimation. In detail, the CNN model showed a better prediction result than RF (CNN: mean error of 4.84% and std. of 7.65%; RF: mean error of 5.92% and std. of 9.77%). On the other hand, in the case of higher SICs (over 95%), both models showed under-estimation. The RF model shows -4.62% of error and 4.57% of standard deviation, but the CNN shows -4.17% and 4.14%, respectively. With the same training samples, the CNN resulted in higher prediction accuracy on both lower and higher SICs. It might be because of the larger learning capacity of CNN than RF (Wang et al., 2017).

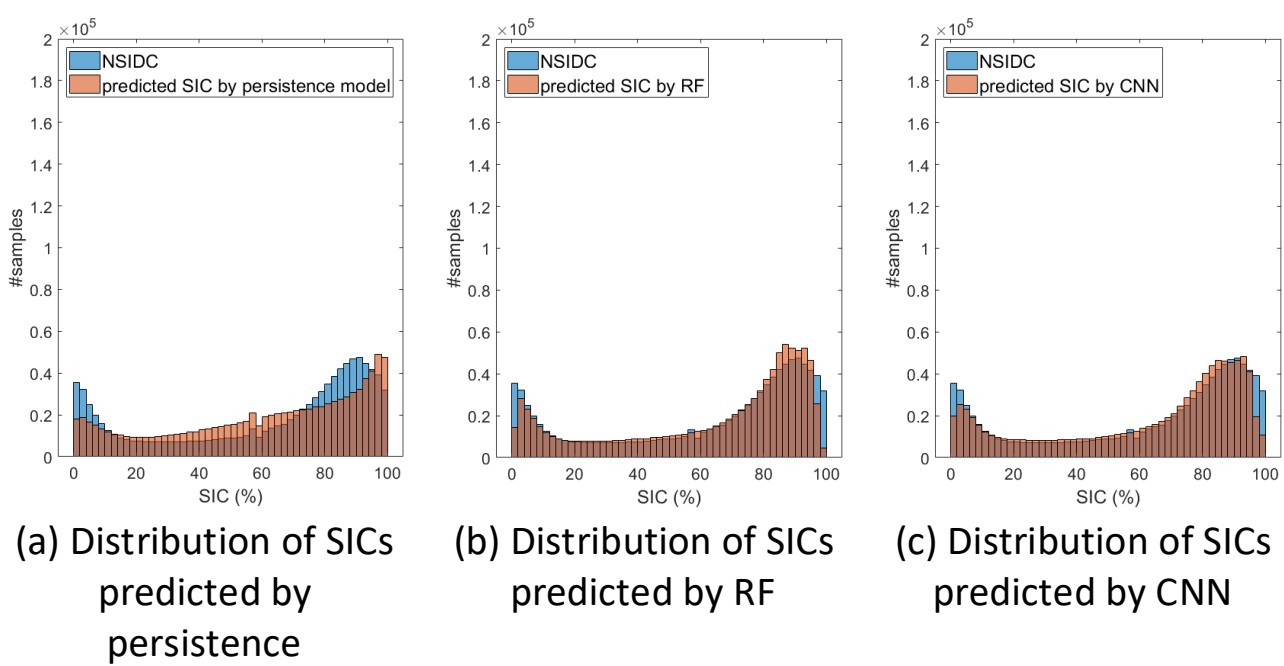

(a) Distribution of SICs predicted by persistence

(b) Distribution of SICs predicted by RF

(c) Distribution of SICs predicted by CNN


**Figure 4.** Histograms of SICs based on NSIDC (blue) and three models (red) in the melting season (Jun. – Sep.) during 2000-2017.

The spatial comparison of the predicted September SIEs in 2017 between the reference (NSIDC) and three approaches used
in this study is shown in Figure 5. The observed SIE in Sep. 2017 was 4.80 million km$^2$ which was reported by the Sea Ice
Prediction Network (http://www.arcus.org/sipn). The SIE in Sep. 13, 2017 was the eighth-lowest in the satellite record since
1981 (NSIDC, 2017). The SIEs predicted by the anomaly persistence, RF and CNN models were 4.37, 4.95, and 4.88 million
km$^2$, respectively. While the anomaly persistence model under-estimated the SIE, the other two models slightly over-estimated.
The anomaly persistence model considered the decreasing trends of sea ice somewhat excessively. The CNN-based model
showed the lowest prediction error compared to the Sea Ice Prediction Network reference data (0.09 million km$^2$). In terms of
spatial distributions, the anomaly persistence model showed the excessive retreat of sea ice in the Beaufort and Laptev Sea
(Fig. 5a). However, the RF and CNN models showed slightly wide SIE in the Chukchi and Barents Sea (Figs. 5b and c). The
over-estimated SIE might be because of the July storm across the central Arctic Ocean through the Barents Sea (West and
Blockley, 2017). The accuracy of one-month SIE prediction based on three approaches was compared to the other 37 SIO
contributions for Sep. 2017 (Fig. 5d). Since the SIO reports contain only quantitative SIE values, it was not possible to compare
their spatial distributions. With regard to the SIE values, the statistical approaches showed quite accurate prediction results
based on Arctic sea ice thickness distributions and ice velocity data (UTokyo) and non-parametric statistical model
(Slater/Barrett NSIDC). The CNN prediction result showed relatively accurate prediction accuracy.

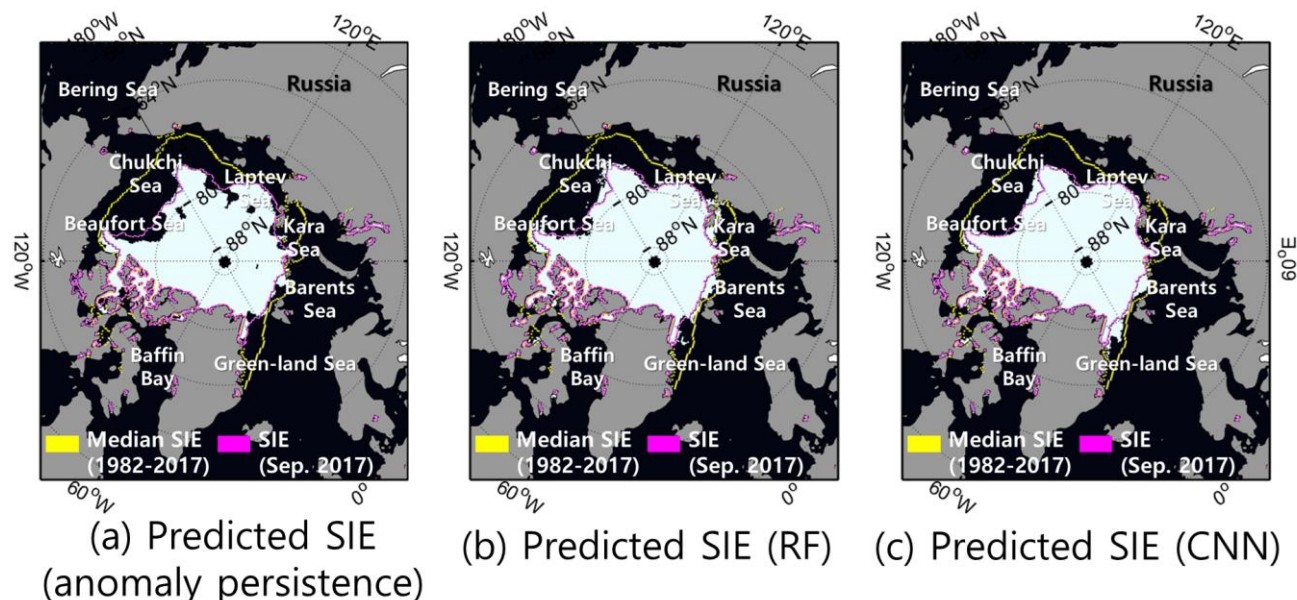

(a) Predicted SIE (anomaly persistence)

(b) Predicted SIE (RF)

(c) Predicted SIE (CNN)

(d) Distribution of SIOs for September 2017 extent (August report)

**Method type**

Heuristic | Statistical | Mixed | Dynamical | This study

**Figure 5.** The predicted SIEs using the anomaly persistence (a), RF (b), and CNN (c) for Sep. 2017. Distribution of SIO values for Sep. 2017 SIEs reported in Aug. 2017. (d).


Since the persistence model did not work well when compared to the RF and CNN models, the subsequent analyses are focused on the RF and CNN models. Figure 6 shows monthly prediction accuracies (i.e., RMSE and nRMSE) for the RF and the CNN

models. The RF model showed lower prediction accuracy than the CNN model for all months. With regards to the RMSE of the CNN model, the prediction accuracy was higher in the melting season (Jun. – Sep.; 5.41%) than in the freezing season (Dec. – Mar.; 6.13%). However, as mentioned, the RMSE considers the range of sample values; for instance, more zero or low SIC values were found in the melting season (Chi and Kim, 2017). Thus, the nRMSE showed the opposite pattern to the RMSE. The normalized RMSE using the standard deviation can show the prediction accuracy considering the different ranges of SIC

by month. In nRMSE of the CNN model, there is a different pattern between the melting season (Jun. – Sep.; 19.09%) and freezing season (Dec. – Mar.; 14.08%). According to the two-sample t-test, the nRMSE in the melting season is higher than in the freezing season ($p < 0.01$; n = 18) throughout the entire period (2000-2017). The difficulty of SIC prediction in the melting season is a well-known problem because of the unexpected decline of Arctic sea ice in recent years (Stroeve et al., 2007; Chi and Kim, 2017).

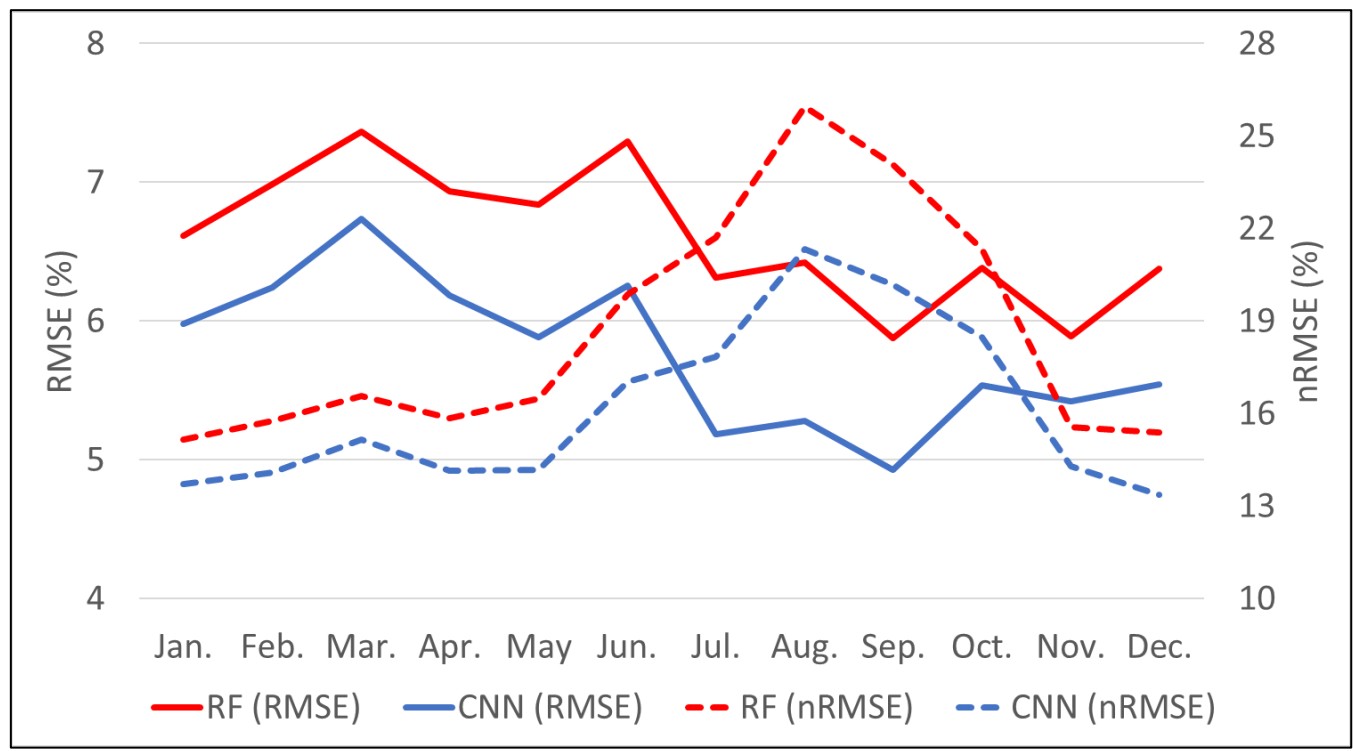


**Figure 6.** Monthly prediction accuracies with differences between two models for the entire periods (2000-2017, RMSEs and nRMSEs).

By focusing on the different patterns of prediction accuracy in the freezing (Dec. – Mar.; nRMSE of 14.08%) and melting season (Jun. – Sep.; nRMSE of 19.09%), the yearly trends in the prediction accuracy of the CNN model were examined (Fig. 7). The nRMSE in the melting season showed an increasing trend in more recent years (2000-2017). Since the dynamic changes

in the Arctic environment, including warm air temperature (Hassol, 2004; Zhang et al., 2007), thinning sea ice (Maslanik et al., 2007), higher ocean surface temperature (Steele et al., 2008) have intensified in recent years, it makes the prediction of SIC in the melting season much more challenging. For instance, the Arctic sea ice extent experienced two major plummets, one in summer 2007, and one in summer 2012 because of multiple causes, such as the unexpected warm atmospheric conditions, radiation anomalies, and summer cyclones (Kauker et al., 2009; Kay et al., 2008; Parkinson and Comiso, 2012; Zhang et al.,
365 2013).

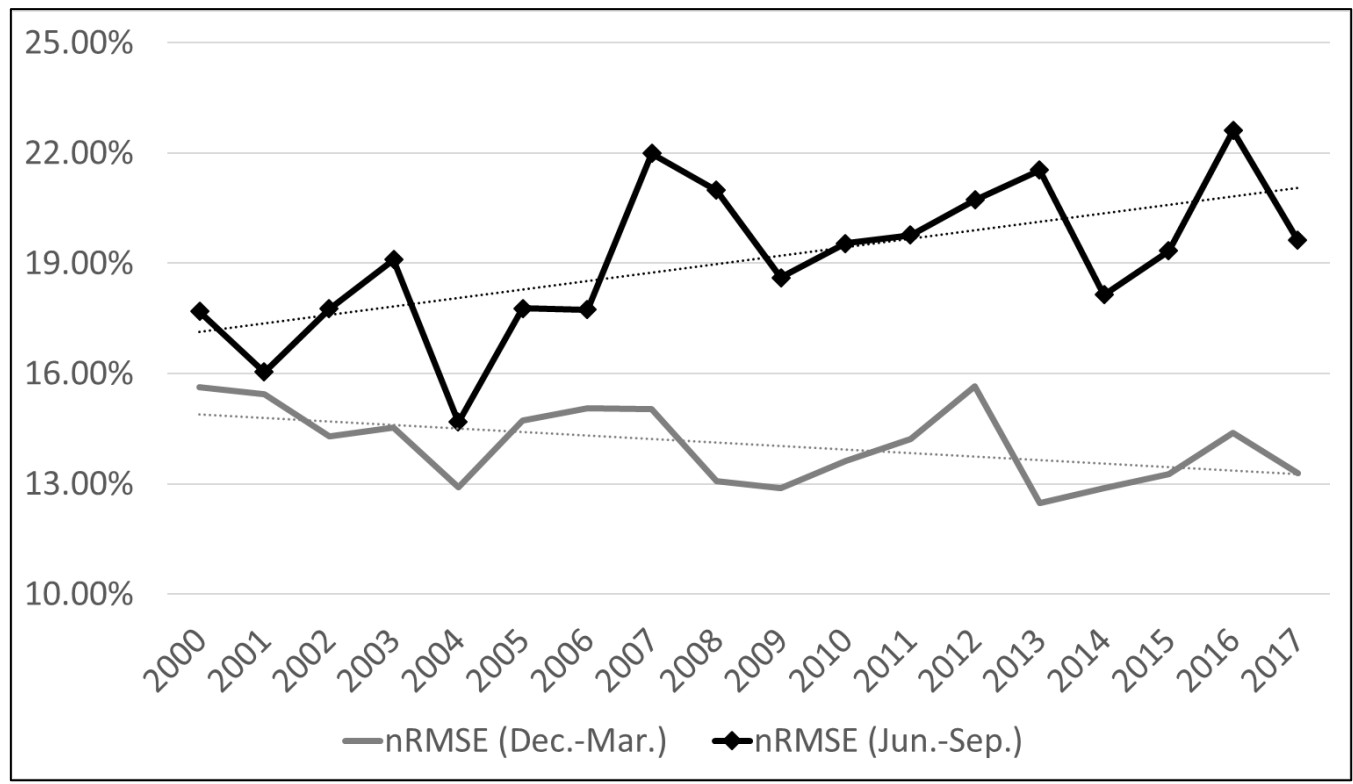

**Figure 7.** Changes of prediction accuracy (nRMSE) using CNN model in freezing (Dec.-Mar.) and melting (Jun.-Sep.) season (2000-2017, dotted lines show trend).

### 4.2 Prediction results in extreme cases: September 2007 and 2012

SIC prediction results of the actual SIC and the SICs predicted by the RF and CNN models were conducted using two extreme cases: September 2007 and 2012 (Fig. 8 and 9). Even though there were unpredicted plummets in the extent of the sea ice, the CNN model showed relatively good prediction results in Sep. 2007 and 2012 (RMSE of 5.00 % and 4.71%, nRMSE of 21.93% and 23.95%, respectively).

In the case of Sep. 2007, there were large sea ice losses through the Beaufort Sea – Chukchi Sea – Laptev Sea during summer
(Fig. 8d). Both the RF and CNN models showed an over-estimation of SIC over the Chukchi Sea and Laptev Sea. This implies
that both models were not able to effectively learn the speed of the drastic retreat of sea ice in that region through training (Fig.
8e-f). Similarly, Fig. 9 shows the prediction results and errors based on the RF and the CNN models in Sep. 2012. In summer
2012, there was also a large loss of sea ice over the Beaufort Sea – Laptev Sea – Kara Sea (Fig. 9d). Both the RF and CNN
models yielded over-estimations of SIC in the region between the Barents Sea and the Kara Sea. This might have been caused
by the fast decline of sea ice in that region because of warm seawater inflows from the Atlantic Ocean in the summer season
(Schauer et al., 2002; Årthun et al., 2012; Kim et al., 2018). The results of two extreme cases showed that the prediction errors
were mainly found in the regions that show high SIC anomalies (i.e., marginal ice zone with small training sample size; Figs.
8d-f and 9d-f).

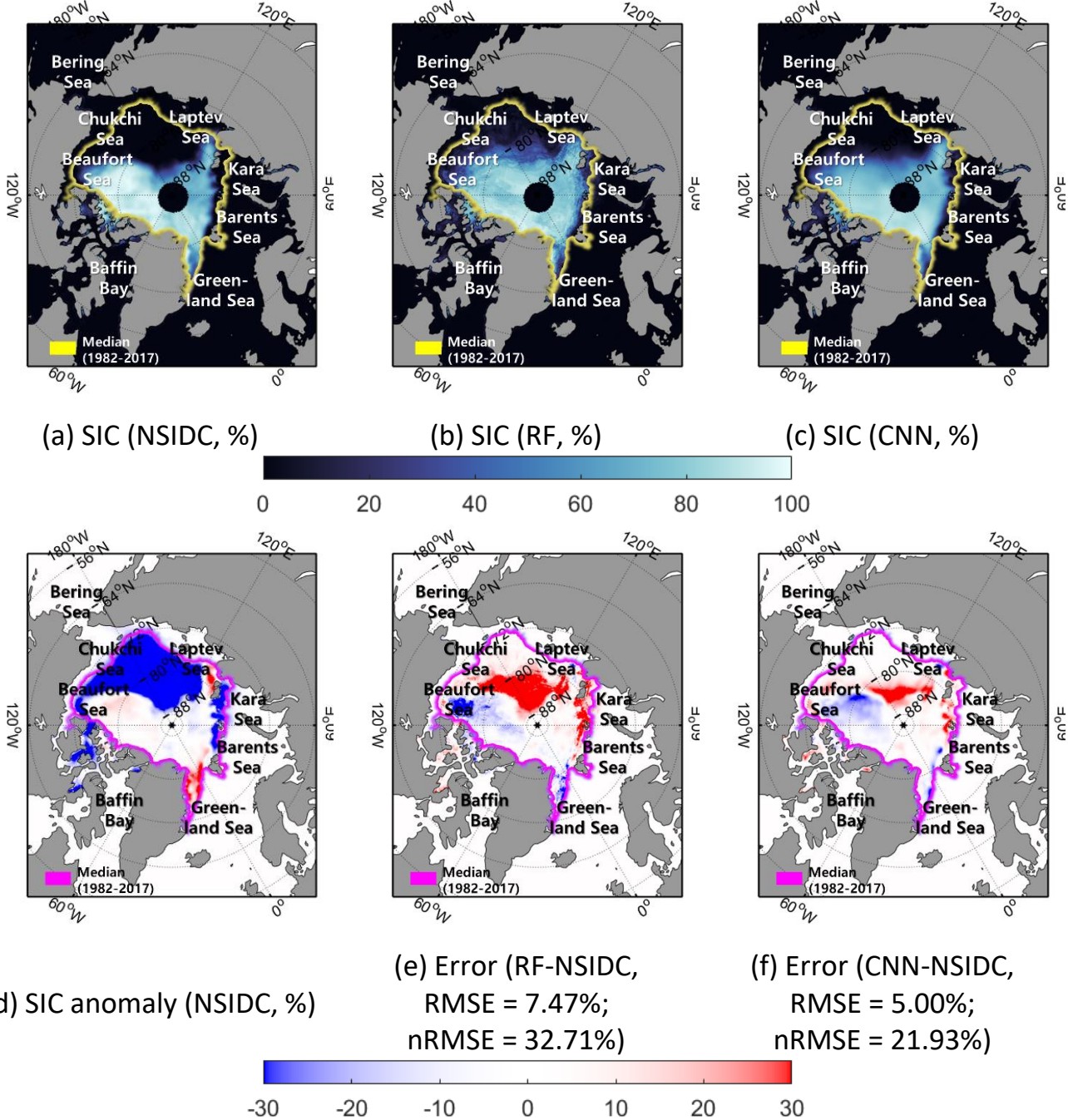

(a) SIC (NSIDC, %)  (b) SIC (RF, %)  (c) SIC (CNN, %)

(d) SIC anomaly (NSIDC, %)

(e) Error (RF-NSIDC,
RMSE = 7.47%;
nRMSE = 32.71%)

(f) Error (CNN-NSIDC,
RMSE = 5.00%;
nRMSE = 21.93%)

**Figure 8.** The actual SIC (a), predicted SICs (b-c), SIC anomaly (d), and errors between predicted and the actual SICs (e-f) in September 2007.

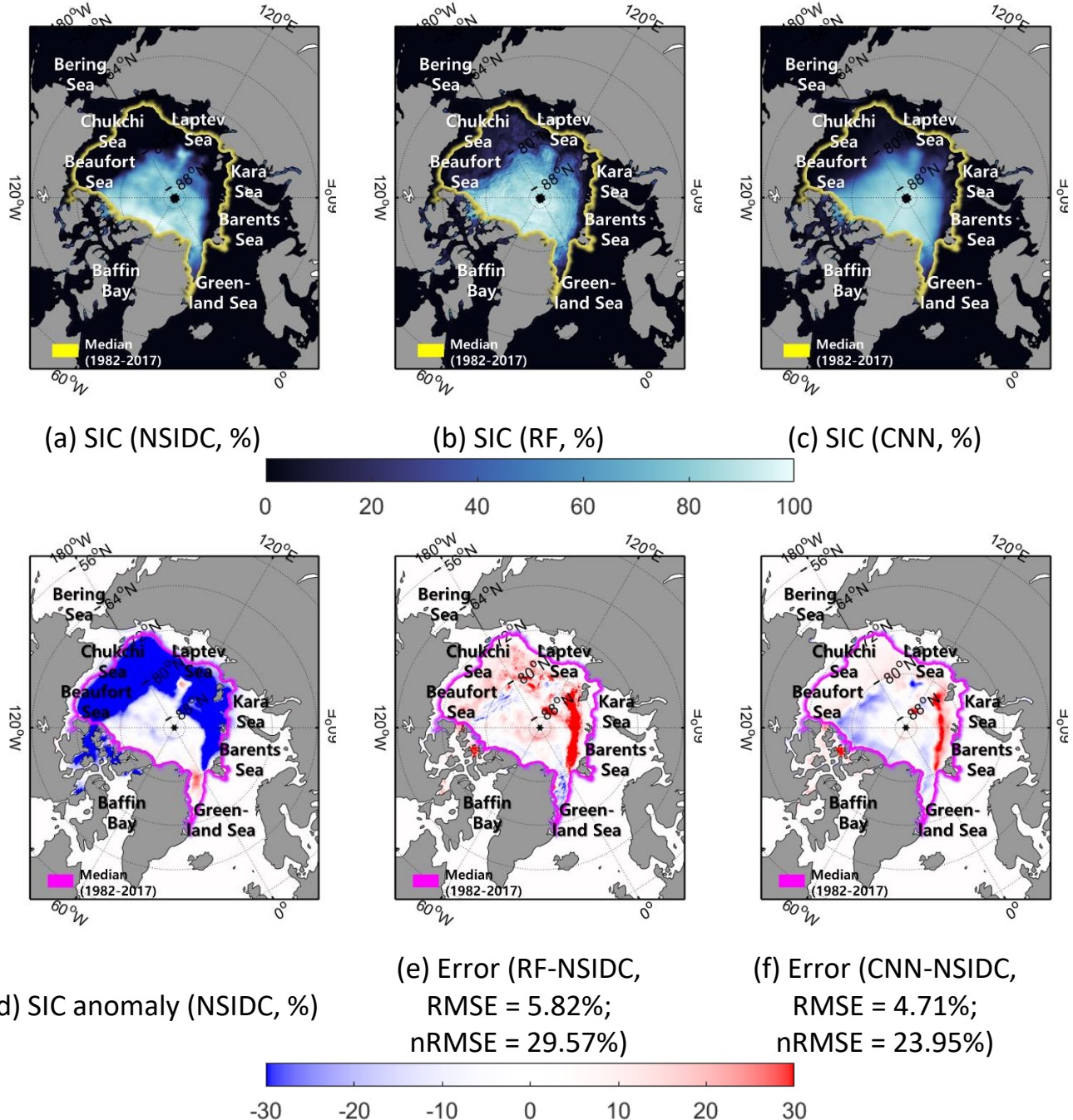

**Figure 9.** The actual SIC (a), predicted SICs (b-c), SIC anomaly (d), and errors between predicted and the actual SICs (e-f) in September 2012.


Together, Figs. 10 and 11 show a detailed analysis focusing on the regions containing high numbers of prediction errors in September 2007 and 2012. Interestingly in both cases, over-estimation was found in no ice zones directly neighboring the marginal sea ice zone (dotted black circle area, Figs. 10 and 11c-d). Both cases show high SST and T2m anomalies together with a low FAL anomaly, caused by a melted snow layer (Figs. 10 and 11i-k). Those anomalous patterns of SST, T2m, and

FAL were caused by anomalous strong solar radiation for both cases (Kauker et al., 2009; Kay et al., 2008; Parkinson and Comiso, 2012; Zhang et al., 2013). In regards to v-wind, the anomalous warm wind toward the Arctic center, inflowed by strong southerly winds driven from the Pacific water, resulted in melting in the Beaufort Sea in 2007 (Zhang et al., 2008, Fig. 10l). However, the CNN model did not catch the past negative SIC anomalies effectively. For instance, Figs. 10d and h depict overestimation errors in the northern part of the region by showing negative SIC anomalies. Similarly, Figs. 11d, g, and h

document over-estimations in the northern part of the region that shows negative SIC anomalies near the Barents Sea and the Kara Sea. Such over-estimation might be caused by the use of a small moving window (i.e., 11-by-11). Since the anomalies were found quite far from the marginal sea ice zone, the models were not able to predict changes in sea ice well. However, a larger window size might impede the overall performance of the model by forcing it to deal with too much learnable information in the CNN approach (Lai et al., 2015). A detailed exploration of the optimum window size is needed in future

research.

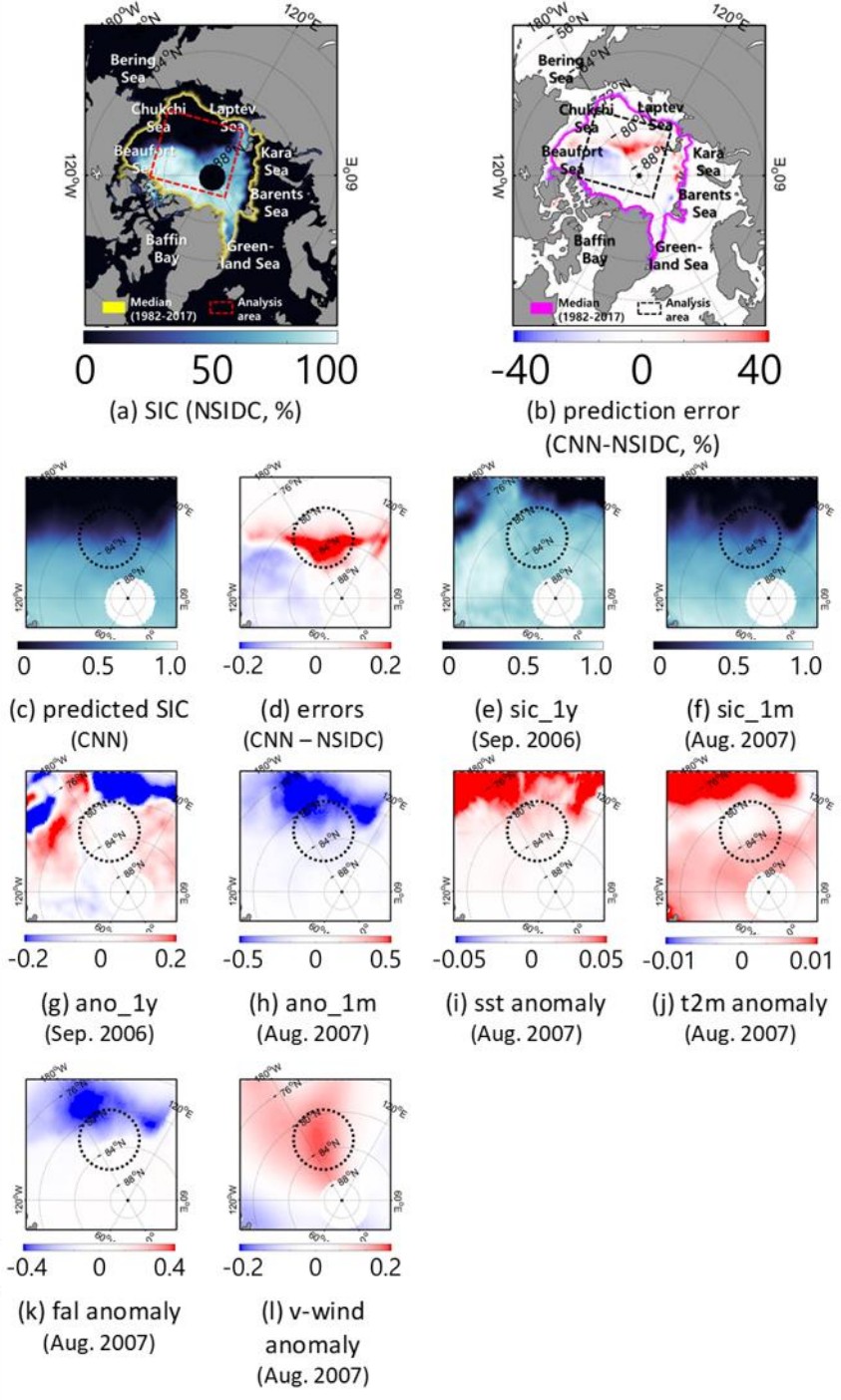

**Figure 10.** Comparison of the prediction results of both models with eight input variables in the Beaufort Sea–Laptev Sea in September 2007. Dotted black circle: the region shows higher prediction errors.

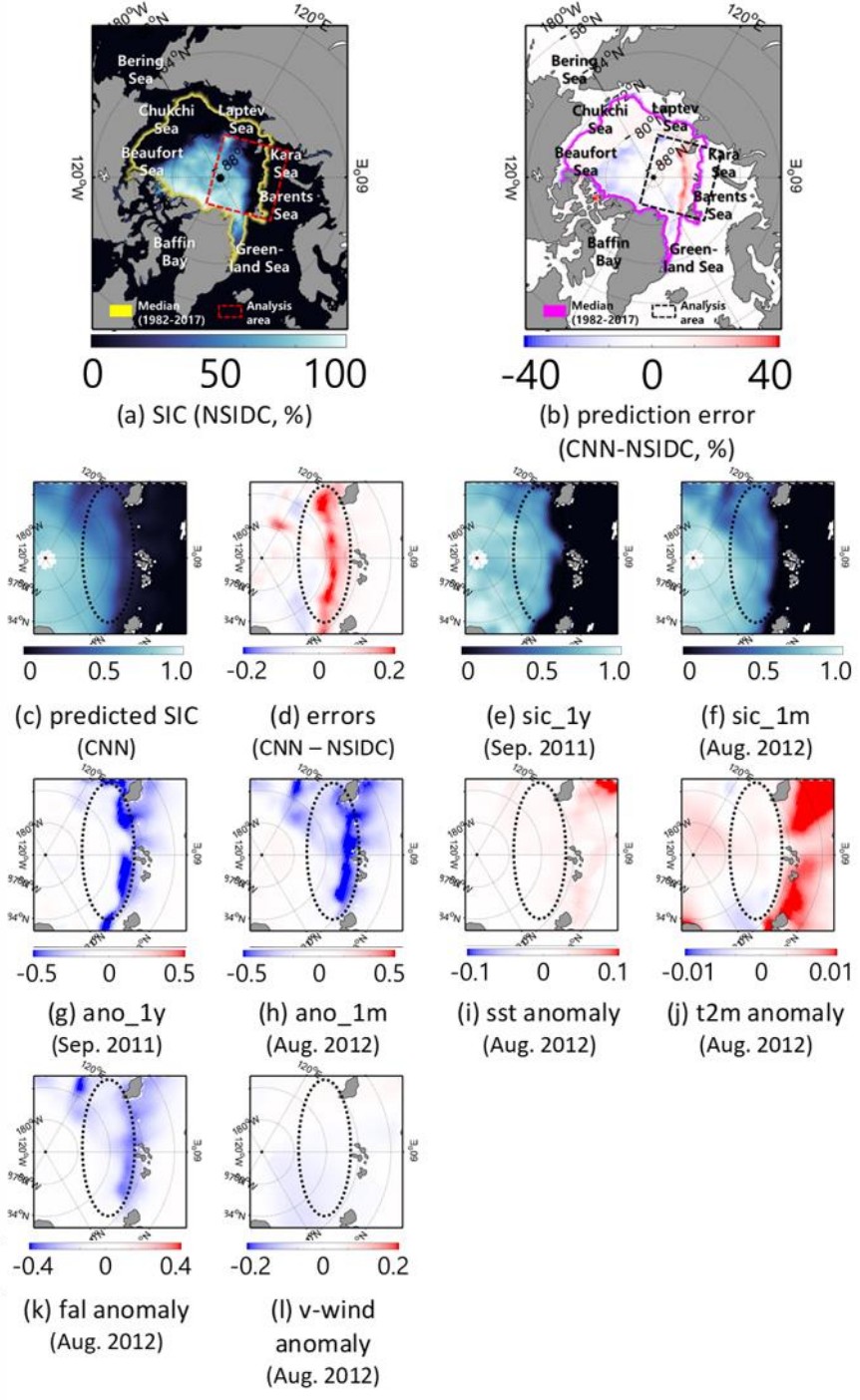

**Figure 11.** Comparison of the prediction results of both models with eight input variables in the Barents Sea –Kara Sea in September 2012. Dotted black circle: the region shows higher prediction errors.

### 4.3 Variable sensitivity

Table 3 shows the variable sensitivity results of both models from 2000 to 2017. The two models show SIC-related variables as the most sensitive factor, i.e. sic_1m and sic_1y, rather than other oceanic or climate variables. These results are consistent

for each model in the annual mean, freezing season (Dec.-Mar.), and melting season (Jun.-Sep.). As the SIC-related variables have a role regarding the time-series climatology information of sea ice, SICs themselves can affect SIC prediction in the future (Deser and Teng, 2008; Chi and Kim, 2017). Between long-term climatologies (sic_1y and ano_1y) and short-term climatologies (sic_1m and ano_1m), the former showed higher sensitivity in both models (except sic_1y and sic_1m in the RF). The previous studies have revealed the clear yearly sea ice trends of each month by investigating monthly averaged sea

ice extents of the nine Arctic regions and the total from 1979 (Cavalieri and Parkinson, 2012; Parkinson and Cavalieri, 2002). Thus, the monthly models showed long-term climatologies as more contributing factors than the other variables (i.e., SICs in past Jan. is important in the Jan. prediction model). Although long-term climatologies were important in the monthly models, the RF model identified sic_1m as the most contributing factor than sic_1y. It might be due to the limitation of the input variables of the RF model used in this study, resulting in a lack of detailed spatial information. The RF model considered

spatial information based on 24 additional proxies using an 11-by-11 window (i.e., mean, minimum, and maximum). However, it may not be sufficient to examine the various spatial distributions of input variables. As a result, the RF model might be highly influenced by short-term information rather than long-term variables.

**Table 1.** The average variable sensitivity for the RF and CNN models during 2000-2017 (annual mean, freezing season (Dec.-Mar.), and melting season (Jun.-Sep.)).

|  |  | sic_1y | sic_1m | ano_1y | ano_1m | SST | T2m | FAL | v-wind |
|---|---|---|---|---|---|---|---|---|---|
| RF | Annual mean | 1.098 | **1.107** | 1.086 | 1.032 | 1.059 | 1.029 | 1.080 | 1.018 |
| RF | Freezing season | 1.080 | **1.091** | 1.087 | 1.045 | 1.053 | 1.011 | 1.071 | 1.019 |
| RF | Melting season | 1.098 | **1.104** | 1.099 | 1.031 | 1.045 | 1.060 | 1.079 | 1.034 |
| CNN | Annual mean | **1.134** | 1.029 | 1.095 | 1.012 | 1.035 | 1.005 | 1.006 | 1.008 |
| CNN | Freezing season | **1.145** | 1.063 | 1.113 | 1.026 | 1.042 | 1.024 | 1.015 | 1.026 |
| CNN | Melting season | **1.121** | 1.033 | 1.090 | 1.017 | 1.054 | 1.010 | 1.005 | 1.015 |

*The highest value is highlighted*

**4.4 Variable sensitivity in extreme case: September 2007 and 2012**

Table 4 shows the variable sensitivity focusing on every September in 2000-2017, 2007, and 2012. Unlike the results in Table 3, T2m and FAL were identified as the most influencing factors in the RF model. As reported in many studies, solar radiation has a large effect on the changes in sea ice (Kang et al., 2014; Guemas et al., 2016). In addition, the ice-albedo feedback

contributes to the recovery of sea ice from the losses in summer (Comiso, 2006; Tietsche et al., 2011). In the case of September
2007, the warm surface air temperature was the main cause of the drastic decrease of sea ice (Kauker et al., 2009). However,
in the case of v-wind, a Gaussian noise made an improvement to the prediction accuracy in two extreme cases for the RF
model. While there are no studies revealing the effects of v-wind in Sep. 2012, there is an indirect effect from the southerly
warm wind toward the Arctic center in Sep. 2007 (Zhang et al., 2008). Moreover, in the RF model, the degree of sensitivity of
FAL is bigger in the two extreme cases than for the entire period. These pieces of evidence may point out that the RF model
is less robust than the CNN model to highly anomalous SIC cases. In contrast to the RF model, the CNN model consistently
identified the sic_1y as the most contributing variable. Although there is no clear causality between the SICs one-year before
and the anomalous decline of sea ice in Sep. 2007 and 2012, past SICs provide information on SICs in the future as time-series
data (Chi and Kim, 2017).

**Table 2.** The average relative variable importance for the RF and CNN models in September (2000-2017 average, 2007, and 2012).

| | | sic_1y | sic_1m | ano_1y | ano_1m | SST | T2m | FAL | v-wind |
|---|---|---|---|---|---|---|---|---|---|
| RF | Average | 1.095 | 1.069 | 1.137 | 1.067 | 1.072 | 1.148 | **1.165** | 1.070 |
| | 2007 | 1.136 | 1.122 | 1.177 | 1.118 | 1.225 | **1.258** | 1.207 | 0.996 |
| | 2012 | 1.126 | 1.057 | 1.102 | 1.064 | 1.096 | 1.100 | **1.207** | 0.997 |
| CNN | Average | **1.090** | 1.035 | 1.056 | 1.005 | 1.009 | 1.000 | 1.002 | 1.004 |
| | 2007 | **1.133** | 1.046 | 1.091 | 1.022 | 1.017 | 1.007 | 1.008 | 1.015 |
| | 2012 | **1.078** | 1.054 | 1.041 | 1.020 | 1.040 | 1.034 | 1.023 | 1.028 |

*The highest value is highlighted*

Figure 12 shows the spatial influence of two sets of variables with impulse noise (zero values). As shown in Figs. 12b and e,
the CNN model was not able to predict SICs in the existing sea ice area when using zero values for the SIC-related variables
(sic_1y, sic_1m, ano_1y, and ano_1m). When the CNN model set zero values for the other environmental variables (SST,
T2m, FAL, and v-wind), the model was not able to predict a decrease of SICs around the marginal areas between the sea ice
and open sea (Fig. 12c and f). It is possibly due to decays on the marginal ice zone by anomalous SST, T2m, and FAL in both
cases. Consistent with the results of the sensitivity analysis (Table 4), SIC-related variables were identified as important
indicators to predict SICs (Deser and Teng, 2008). The other meteorological and oceanographic variables tended to affect the
SIC changes of the marginal zone ice, particularly, the neighboring thin ice and no ice zone (Stroeve et al., 2008; Chevallier
et al., 2013; Zhang et al., 2013).

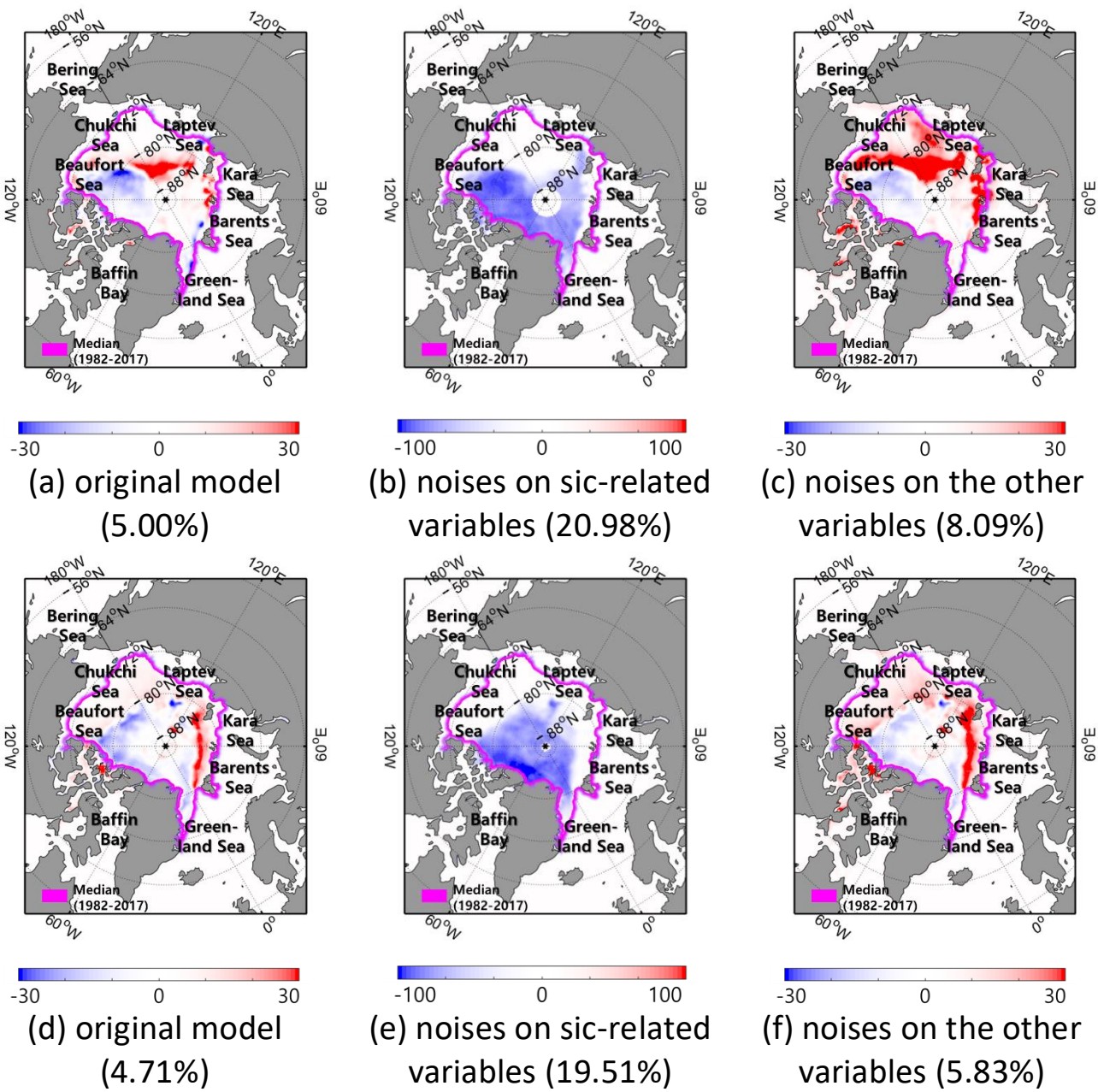

455 **Figure 12.** The prediction errors (predictions by CNN – NSIDC, %) and RMSE (%) from three prediction results in (a-c) September 2007 (d-f) and 2012: (a and d) original model, (b and e) with noises on SIC variables (sic_1y, sic_1m, ano_1y, and ano_1m), (c and f) with noises on the other variables.

### 4.5 Novelty and limitations

Our study developed a novel one-month SIC prediction model using the CNN deep learning approach. The research findings

460 from this study can make a contribution towards filling the gaps in the research on short-term sea ice change and prediction

using a deep-learning approach (Grumbine, 1998; Preller and Posey, 1989). Our short-term SIC prediction model can provide valuable information, which can be used in various decision-making processes in the maritime industry and in research regarding sea ice forecasting (Schweiger and Zhang, 2015). Notably, the non-linear learning architectures of the CNN model showed good prediction accuracy based on the larger learning capacity and more consistent temporal SIC prediction than the traditional machine learning approach (Wang et al., 2016; Liu et al., 2018).

However, there are some challenging limitations to the proposed CNN model, particularly regarding the prediction variables. First, this study did not consider the effects of a longer time scale, or persistent effects, on sea ice changes (Guemas et al., 2014). For example, the 2007 and 2012 sea ice minimums were caused by not only the anomalous warm atmospheric conditions of the summer season but also by persistently warm winter and spring seasons, which especially affected the melting in the marginal ice zone (Devasthale et al., 2013). The proposed CNN model could be used for the longer prediction (i.e., two- or three-month prediction) in consideration of the persistent effects of input variables such as SST and T2m. Moreover, additional input variables that represent seasonal, or longer-term variabilities of the Arctic environment should be considered in the proposed models. The persistence of sea-ice volume and atmospheric circulation related variables would be suitable for the long-term sea ice forecast (Guemas et al., 2014). Second, the sea ice thickness is an important factor when predicting sea ice changes because the thinner sea ice is relatively vulnerable to melt (Stroeve et al., 2008; Chevallier et al., 2013; Zhang et al., 2013). However, we did not consider sea ice thickness data because of the limited availability of reliable sea ice thickness products. Third, there is a well-known problem with deep-learning models — interpretability. Because of complicated and non-linear connections between hidden layers, the deep learning models are hard to interpret (Koh et al., 2017; Guidotti et al., 2018). Recent deep learning studies have attempted to report explainable results using various visualization approaches such as heat maps and occlusion maps (Brahimi et al., 2017; Trigueros et al., 2018). The present study explained the model using a variable sensitivity analysis, as well as the inspection of the spatial distribution. However, the model still has problems providing clear interpretations of the non-linear relationships among variables.

## 5. Conclusion

The main purpose of this study was to develop a novel one-month SIC prediction model using the CNN approach. The CNN model showed better prediction performance (MAE of 2.28%, ACC of 0.98, RMSE of 5.76%, nRMSE of 16.15%, and NSE of 0.97) than the persistence forecast (MAE of 4.31%, ACC of 0.95, RMSE of 10.54%, nRMSE of 29.17%, and NSE of 0.89) and RF models (MAE of 2.45%, ACC of 0.98, RMSE of 6.61%, nRMSE of 18.64%, and NSE of 0.96). The prediction accuracy in the melting season (Jun. – Sep., nRMSE of 19.09%) was lower than the freezing season (Dec. – Mar., nRMSE of 14.08%). The overall prediction accuracy decreased in the more recent years because of the accelerated sea ice melting caused by global warming. In two extreme cases, the CNN model yielded promising prediction results with respect to RMSE, as well as the spatial distribution of SICs (less than 5% RMSE). The prediction errors normally occurred in the marginal ice zone, which has

higher sea ice anomalies. From the variable sensitivity analysis using CNN, the SICs one-year before was identified as the most important factor in predicting sea ice changes. While the SIC-related variables had high effects on SIC prediction over ice-covered areas, the other meteorological and oceanographic variables were more sensitive in predicting the SICs in marginal
ice zones.

*Data availability.* The research data can be obtained by request to the corresponding author (ersgis@unist.ac.kr).

*Author Contribution.* Y.K. led manuscript writing and contributed to data analysis and research design. H.K. and S.L. contributed to the research design and discussion of the results. D.H. contributed to data processing and analysis. J.I supervised
this study, contributed to the research design, manuscript writing and discussion of the results, and served as the corresponding author.

*Competing interests.* The authors declare that they have no conflict of interest.

**Acknowledgement**

This study was supported by the Korea Polar Research Institute (KOPRI) Grant PE19900 (Development of algorithms to
extract satellite-based Arctic sea ice characteristics), the Korea Meteorological Administration (KMIPA 2017-7010), and the National Research Foundation of Korea (NRF- 2017M1A3A3A02015981).

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
