# Peer review of "Prediction of monthly Arctic sea ice concentrations using satellite and reanalysis data based on convolutional neural networks"

_The Cryosphere, 2019_

## Referee Comment (RC1) · Anonymous Referee #1 · 3 Oct 2019

Main comments:

(1) On the choices of SIC predictors, I think more justification should be provided regarding why all of these predictors are needed. I would argue that they are not.

From my understanding of the model and intuition of the problem, the SIC information from the year before should only be contributing information to the statistical model regarding the trend in observed SIC – i.e. there's no physical explanation why these interannual variations would contribute to skill one year in advance. However, sic_1y and ano_1y both contain interannual variability information as well. Those fluctuations are just noise in this case and very likely resulting in over-fitting. I would suggest

replacing both of these predictors simply by the observed trend, and see how this affects the model results. If they are kept, these issues should be addressed.

Second, what is the reasoning behind using both sic_1m and ano_1m? Apart from the baseline used to compute anom_1m, they should be perfectly correlated with one another, which introduces the problem of multicollinearity. Have the authors tried only using anom_1m and not sic_1m? Again, if both are kept these issues should be addressed.

(2) On the simple statistical model as reference:

The use of a yearly trend extrapolation as a reference forecast is a bit conservative for the one-month lead time forecast, since there is typically high autocorrelation at a lag of one month for SIC (even after accounting for the long-term trend). A more robust reference, and what is commonly used, is an anomaly persistence forecast, or a damped anomaly persistence forecast (e.g. see Wang et al 2016). Ideally this would be done by persisting anom_1m one month ahead and adding it to the observed trend calculated for that month (as opposed to climatology as done in Wang et al 2016).

(3) Fig. 2 - it would be helpful to split these up for the melt and freeze seasons. Showing the annual mean makes it difficult to interpret the figure.

(4) Why are only error/bias metrics considered? Include a figure analogous to Fig. 2 showing maps of anomaly correlation for the de-trended predictions relative to de-trended obs would provide a more comprehensive analysis.

(5) The paper needs to be reviewed carefully for proper use of the english language. There are several spelling and grammatical errors.

Minor comments: L35-40: "Arctic sea ice has been rapidly declining, which impacts not only the Arctic climate, but also mid-latitudes (Yu et al., 2017)" should say "... but also possibly the mid-latitudes". There is not yet a consensus on this matter.

L40: Usually we say "projecting climate change"; "forecasting" refers to an initial-value

problem.

L 40-45: Should reference Drobot et al, 2003, 2006; Lindsay et al. 2008; and Wang et al. 2016 for statistical predictions of sea ice concentration.

L55-60: "Studies regarding short-term sea ice forecasting have received relatively little attention (Grumbine, 1998; Preller and Posey, 1989)." Specify that you're referring to machine learning predictions; studies have considered this with classic statistics models (e.g. references on previous comment).

L60: "SIC describes the amount of the sea that is covered by ice". 'The sea' is poorly defined; better to say "SIC describes the fraction of a specified area (typically a grid cell) covered by sea ice".

L185: eqs 1-4; please expand on how these error metrics are computed with respect to space and time. I would think that the spatial averaging would have to be done before any temporal averaging in order for the mask (based on the previous nine years w.r.t the forecast month) to be applied effectively... Is this correct?

L250: "The model did not catch well the decreasing trends of sea ice due to global warming." The model is a linear trend, so that's exactly what it should be doing unless the actual trend is accelerating. I would think It's more likely that the trend forecast is showing 100% SIC in the central Arctic in years like 2007 and 2012 when sea ice retreated further north than the marginal ice zone predicted by the trend. However, the fact that high-SIC values that are less than 100% do not show this bias in Fig. 3a is highly suspect. Are the authors sure an error hasn't been made either making the figure or in calculating the reference forecast?

Fig. 5: Was the impulse noise (i.e. setting to zero) used on the predictors over the training sample and the testing (i.e. forecast) sample? Are the predictions for 2007 and 2012 with impulses on the SIC variables (panels b and c) just that there is no sea ice (all very negative values)? If the impulses are applied over the training data, it's

[Figure]

hard to imagine why the remaining variables wouldn't be capable of "creating ice" in the model, even if it's just a climatology.

References: Drobot, S. (2003). Long-range statistical forecasting of ice severity in the Beaufort–Chukchi Sea. Weather and Forecasting, 18(6), 1161-1176.

Drobot, S. D., Maslanik, J. A., & Fowler, C. (2006). A long‐range forecast of Arctic summer sea‐ice minimum extent. Geophysical Research Letters, 33(10).

Wang, L., Yuan, X., Ting, M., & Li, C. (2016). Predicting summer Arctic sea ice concentration intraseasonal variability using a vector autoregressive model. Journal of Climate, 29(4), 1529-1543.

Lindsay, R. W., Zhang, J., Schweiger, A. J., & Steele, M. A. (2008). Seasonal predictions of ice extent in the Arctic Ocean. Journal of Geophysical Research: Oceans, 113(C2).

---

## Referee Comment (RC2) · Anonymous Referee #2 · 14 Oct 2019

Review of

**Prediction of monthly Arctic sea ice concentration using satellite and reanalysis data based on convolutional neural networks**
by Kim et al.

Manuscript number: tc-2019-159

**General comments:**

The paper presents a new one-month sea ice concentration prediction model using the Convolutional Neural Network deep learning approach. Output is compared to the results of a Random Forest and a simple prediction model. Models are applied to the time period from 1988 to 2017 and extreme cases of sea ice concentration decline are analysed in detail.

The subject is appropriate for TC. The title reflects the content of the paper, the abstract provides a complete summary and the paper is generally well structured. The review of existing published work is good, the number of references is appropriate.

Overall, figures and tables are clear and their captions self-explanatory. However, few figures should by improved according to specific comments below.

Especially the selection of predictors is not convincing and should be justified in more detail. Regarding the atmospheric predictors, why is FAL and v-wind necessary?

Why is a simple linear extrapolation model used for a one-month prediction?

**Specific comments:**

Figure 1: A larger font should be used in the bottom right part of the figure.

Figure 2: Results should be shown for the freeze and the melt season separately.

Figure 8, Figure 9: A larger font should be used.

**Technical corrections:**

There are numerous spelling and grammatical errors in the text, which should be eliminated.

---

## Author Comment (AC1) · 15 Nov 2019

The authors would like to thank the referees for their precious time and invaluable comments. The corresponding changes and refinements are highlighted in yellow in the revised paper and are also summarized in our responses below. Authors' responses are in blue. Reviewer's comments are in black. When the manuscript in cited, it is shown in italics.

**Response to anonymous referee #1**

**Main comments:**
1) On the choices of SIC predictors, I think more justification should be provided regarding why all of these predictors are needed. I would argue that they are not.

➔ We conducted a feature selection process in the early stage of the study. Including the eight predictors, four additional predictors were used for feature selection to develop the CNN model:
- o ice surface temperature (IST), which affects a heat balance that determines the growth or
decay of sea ice (Gabison, 1987; Guemas, 2014);
- o mean sea level pressure (MSL), which is a driving force to make wind variability on the Arctic region as well as sea ice drift (Tsukernik et al., 2009; D¨oscher et al., 2010; Guemas, 2014);
- o total cloud cover (TCC), which is a proxy of the amount of solar radiation like a FAL
(Kay et al., 2008; Kang et al., 2014);
- o and 10-meter u-wind vector (u-wind), which transfers heat energies across to the Arctic region and affects on growth or decay of sea ice (Arfeuille et al., 2000; Guemas, 2014).
➔ Then we selected predictors using mean decrease accuracy (MDA) based on random forest. The MDA has widely used feature selection criteria by measuring the accuracy changes by
randomly permuting input variables (Strobl et al., 2007; Archer and Kimes, 2008).
➔ Finally, we selected the eight predictors based on the mean MDA from twelve monthly prediction models from 1988 to 2017 using the RF model (Supplementary Table 1 below).

Supplementary Table 1. MDA for input variables used for feature selection in random forest

| ano_1y | t2m | ano_1m | fal | sst | v-wind | sic_1y | sic_1m | tcc | u-wind | msl | ist |
|---|---|---|---|---|---|---|---|---|---|---|---|
| 7.32 | 6.93 | 6.09 | 4.26 | 4.17 | 4.02 | 3.51 | 3.29 | 2.89 | 2.45 | 1.78 | 1.24 |

(references)
Gabison, R. (1987). A thermodynamic model of the formation, growth, and decay of first-year sea ice. Journal of Glaciology, 33(113), 105-119.
Guemas, V., Blanchard-Wrigglesworth, E., Chevallier, M., Day, J. J., Déqué, M., Doblas-Reyes, F. J.,
Fučkar, N. S., Germe, A., Hawkins, E., Keeley, S. and others: A review on Arctic sea-ice predictability and prediction on seasonal to decadal time-scales, Q. J. R. Meteorol. Soc., 142(695), 546–561, doi:10.1002/qj.2401, 2016.
Tsukernik M, Deser C, Alexander M, Tomas R. 2009. Atmospheric forcing of Fram Strait sea ice export: A closer look. Clim. Dyn. 35: 1349–1360, doi: 10.1007/s003-82-009-0647-z.
D¨oscher R, Wyser K, Meier M, Qian M, Redler R. 2010. Quantifying Arctic contributions to climate predictability in a regional coupled ocean–ice–atmosphere model. Clim. Dyn. 34: 1157–1176, doi: 10.1007/s00382-009-0567-y.
Kay, J. E., L'Ecuyer, T., Gettelman, A., Stephens, G. and O'Dell, C.: The contribution of cloud and radiation anomalies to the 2007 Arctic sea ice extent minimum, Geophys. Res. Lett., 35(8),
doi:10.1029/2008gl033451, 2008.

Kang, D., Im, J., Lee, M. I., and Quackenbush, L. J.: The MODIS ice surface temperature product as an indicator of sea ice minimum over the Arctic Ocean. Remote Sens. Environ., 152, 99-108., doi.org/10.1016/j.rse.2014.05.012, 2014.
Arfeuille GL, Mysak A, Tremblay LB. 2000. Simulation of the interannual variability in the wind-driven Arctic sea ice cover 1958–1988. Clim. Dyn. 16: 107–121.
Strobl, C., Boulesteix, A. L., Zeileis, A., & Hothorn, T. (2007). Bias in random forest variable importance measures: Illustrations, sources and a solution. BMC bioinformatics, 8(1), 25.
Archer, K. J., & Kimes, R. V. (2008). Empirical characterization of random forest variable importance measures. Computational Statistics & Data Analysis, 52(4), 2249-2260.

➔  We added the justification for the selection of each predictor in the revised manuscript. We also briefly described the feature selection process.

**Lines 86 – 91:** *"In this study, a total of eight predictors were selected and used to predict SIC next*
*month (Table 1) based on the literature and a preliminary statistical analysis of potential predictors through a feature selection process using random forest (Strobl et al., 2007). We selected the eight predictors by comparing the mean decrease accuracy (MDA) changes based on twelve monthly prediction RF models from 1988 to 2017. The MDA has been widely used as feature selection criteria by measuring the accuracy changes by randomly permuting input variables (Archer and Kimes,*
*2008). It should be noted that fewer predictors than the selected eight ones did not produce better results."*

**Lines 108 – 126:** *"The eight predictors selected in this study though random forest-based feature selection have theoretical backgrounds that are related to the characteristics of SIC. First, SIC itself*
*can affect the SIC in the future because it has a clear inter-annual trend through the melting and freezing seasons (Deser and Teng, 2008; Chi and Kim, 2017). It is a useful characteristic when conducting a time-series analysis, and thus, two SIC time-series climatology predictors (SIC one-year before and SIC one-month before) were used in this study. Although there is no physical explanation of why the interannual variations would contribute to the forecasting skill, it clearly worked well in*
*long-term SIC forecasting through previous studies (Wang et al., 2016; Chi and Kim, 2017). Further, we used two supplementary predictors that indicate the anomalies of SIC one-year before and SIC one-month before, in order to consider anomalous sea ice conditions in the models. The anomaly data could give information about SST anomaly along the sea ice edge in terms of the re-emergence mechanism from the melting to the freezing seasons (Guemas et al., 2014). Second, changes in SST*
*and SIC have a significant relationship with each other, with regards to the heat budget (Rayner et al., 2003; Screen and et al., 2013; Prasad et al., 2018). The re-emergence of sea ice anomalies is also partially explained by the persistence of SST anomalies (Guemas et al., 2014). Air temperature and albedo are related to the amount of solar radiation enabling the prediction of SIC changes. The solar radiation heats the surface of the ocean as well as the sea ice. This causes a rise in the SST while also*
*reducing albedo on the sea ice by melting the surface snow or thinning the sea ice (Screen and Simmonds, 2010; Mahajan et al., 2011). Moreover, the surface snow melting produces melt ponds, wet sea-ice surfaces, and wet snow cover (Kern et al., 2016). Warm winds from lower latitudes toward the Arctic can also reduce sea ice (Kang et al., 2014) and local wind forces affect sea ice motion and formation (Shimada et al., 2006). The wind vector also can cause short or long-range sea*
*ice drifts (Guemas et al., 2014), which may influence SIC variation."*

From my understanding of the model and intuition of the problem, the SIC information from the year before should only be contributing information to the statistical model regarding the trend in observed SIC – i.e. there's no physical explanation why these interannual variations would contribute to skill
one year in advance.

➔  On a statistical basis, there is a clear climatological pattern in SICs (Parkinson and Cavalieri, 2002; Deser and Teng, 2008; Chi and Kim, 2017). Although there is no clear physical explanation of why the interannual variations would contribute to the forecasting skill, the
       long-term effects of environmental conditions for SIC variability might be represented by
       SICs one year before.
➜      For example, the different sea ice variability in the Kara Sea and the Barents Sea is
       controlled by underlying environmental conditions (Wang et al., 2016). Further, the
       anomalies of sea ice cause anomalies on SST along the sea ice edge to affect sea ice
conditions (Guemas et al., 2014).

However, sic_1y and ano_1y both contain interannual variability information as well. Those
fluctuations are just noise in this case and very likely resulting in over-fitting. I would suggest
replacing both of these predictors simply by the observed trend, and see how this affects the model
results. If they are kept, these issues should be addressed.

➜      Thanks for the comment. We partially agree with that point. However, we decided to keep
       both predictors based on the following three reasons.
➜      First, they are not highly correlated with each other according to Pearson's correlation
coefficient between SICs and anomalies during the entire study period ($\rho$=-0.39, $p$<0.01).
       Since ano_1y and ano_1m were calculated only for a more recent time period (2001-2017) to
       focus on the recent sea ice changes, they would play different roles in forecasting even
       though they state interannual variability information as well.
➜      Second, their spatial patterns are quite different. They would contribute to different ways in
prediction based on their different spatial patterns in a CNN-based model (Supplementary
       Figure 1).

[Figure]

Left: mean SICs                    Right: mean anomalies

Supplementary Figure 1. Spatial pattern of SICs (left) and SIC anomalies (right).

➜   Third, we attempted to revise the prediction model using only the observed trend by
          removing ano_1y and ano_1m. As a result, the overall prediction accuracy (RMSE)
          significantly dropped from 5.76% to 6.68%.
      ➜   We revised the manuscript to address these issues.

**Lines 101 – 102:** "*Since the anomalies were calculated from the recent years (2001-2017), there is no
      significant multicollinearity issue that could cause overfitting (Pearson's correlation coefficient
      between mean SICs and anomalies ($\rho$) = -0.39, p<0.01).*"

Second, what is the reasoning behind using both sic_1m and ano_1m? Apart from the baseline used to
compute anom_1m, they should be perfectly correlated with one another, which introduces the
problem of multicollinearity. Have the authors tried only using anom_1m and not sic_1m? Again, if
both are kept these issues should be addressed.

➔ The sic_1y and sic_1m were used considering the long and short-term climatology of SICs.
The ano_1y and ano_1m were used considering additional effects on SIC changes due to
other underlying environmental conditions as we responded to the Main Comment (1) above.
➔ As mentioned before, since they were not highly correlated with each other ($\rho$=-0.39,
$p$<0.01), there is no significant multicollinearity issue. In addition, we tested the prediction
model without sic_1m and ano_1m, which resulted in the increase of RMSE from 5.76% to
6.15% and 6.38%, respectively.
➔ We revised the manuscript to address these issues.

**Lines 114 – 117:** *"Further, we used two supplementary predictors that indicate the anomalies of SIC
one-year before and SIC one-month before, in order to consider anomalous sea ice conditions in the
models. The anomaly data could give information about SST anomaly along the sea ice edge in terms
of the re-emergence mechanism from the melting to the freezing seasons (Guemas et al., 2014)."*

2) On the simple statistical model as reference:
The use of a yearly trend extrapolation as a reference forecast is a bit conservative for the one-month
lead time forecast, since there is typically high autocorrelation at a lag of one month for SIC (even
after accounting for the long-term trend). A more robust reference, and what is commonly used, is an
anomaly persistence forecast, or a damped anomaly persistence forecast (e.g. see Wang et al 2016).
Ideally this would be done by persisting anom_1m one month ahead and adding it to the observed
trend calculated for that month (as opposed to climatology as done in Wang et al 2016).

➔ We replaced the simple prediction model to the anomaly persistence forecast model as you
suggested.

**Lines 159 – 160**: *"Finally, an anomaly persistence forecast model was also examined for predicting
the monthly Arctic SIC. The anomaly persistence model is a useful reference for forecast skill for
time-series data (Wang et al., 2016)."*

**Lines 179 – 182**: *"In the case of the anomaly persistence forecast model, the monthly SIC anomaly of
each pixel persisted and the observed trend was calculated for that month ahead. For example, SICs
in Jan. 2000 were predicted by summing one-month persisted anomaly and one-month ahead SIC
from a linear trend of SICs from Jan. 1988 to Dec. 1999 by each grid."*

3) Fig. 2 - it would be helpful to split these up for the melt and freeze seasons. Showing the annual
mean makes it difficult to interpret the figure.

➔ We revised Figure 2 by splitting into the annual mean, melting, and freezing seasons as
suggested. The corresponding text was revised according to Figure 2.

[Figure]

*Figure 2. The mean absolute SIC anomaly (a) and mean absolute errors between predicted SICs and the actual SICs by the persistence (b), RF (c) and CNN (d) during 2000-2017. As in (a) - (d), but for the melting (Jun. – Sep.) and freezing (Dec. – Mar.) season, (e) - (f) and (i) - (l), respectively.*

**Lines 261 – 271**: *"The spatial distribution of the annual MAE of three models from 2000 to 2017 is shown in Fig. 2. From visual inspection, it appeared that the prediction errors were dominant in the marginal areas (i.e., the boundaries between the sea ice and open seas). Since the marginal sea ice, particularly thin ice, is susceptible to change (Stroeve et al., 2008; Chevallier et al., 2013; Zhang et al., 2013), the prediction accuracy may have decreased. Weak predictability on the marginal sea ice zone might be due to a relatively small training sample size over the area. In the melting season, relatively higher prediction errors appeared not only in the marginal area, but also even ice-covered areas near the Arctic center (Fig. 2f-h). On the other hand, in the freezing season, the prediction*

*errors were shown mainly in the marginal area (Fig., 2j-l). Further, relatively higher prediction errors appeared around the Kara Sea and the Barents Sea (Fig. 2a, e, and i). The region from the Kara Sea to the Barents Sea shows consistent sea ice retreats because of inflows of warm and salty*
*ocean water from the Atlantic Ocean into the Barents-Kara Sea (Schauer et al., 2002; Årthun et al., 2012; Kim et al., 2018) and cumulative positive solar radiation in the summer season (Stroeve et al., 2012). Using a visual comparison, it can be seen that the degree of errors is higher in RF than CNN (Fig. 2)."*

4) Why are only error/bias metrics considered? Include a figure analogous to Fig.2 showing maps of anomaly correlation for the de-trended predictions relative to detrended obs would provide a more comprehensive analysis.

➔   We added Figure 3 to show maps of anomaly correlation coefficient (ACC) for the annual
mean, melting, and freezing seasons as suggested. The corresponding text was also added.

[Figure]

Figure 3. The temporal ACC of the persistence (a), RF (b) and CNN (c) during 2000-2017. As in (a) -
(c), but for the melting (Jun. – Sep.) and freezing (Dec. – Mar.) season, (d) - (f) and (g) - (i),
respectively.

Lines 278 – 290: "The spatial distribution of the temporal ACCs of three models from 2000 to 2017 is
shown in Fig. 3. First of all, every prediction model showed quite good skill scores with high positive
correlation (near 1.0, Fig. 3a-c). Interestingly, the ACCs were higher in the marginal area where
showed relatively high prediction errors. Even though the models were weak to predict SIC changes
in the marginal sea ice zone, but they caught decreasing trends of SICs relatively well. On the other
hand, the region near the Arctic center showed relatively low ACCs. In contrast to the marginal sea
ice zone, the Arctic center region is relatively stable to the changes (Stroeve et al., 2008; Chevallier et
al., 2013). Since SICs in the center is almost saturated (100% of SIC) and very stable, it might cause

*lower ACC values even there were relatively small prediction errors. In case of the melting season (Jun. – Sep., Fig. 3d-f), the degree of ACCs decreased when compared to the annual-mean (Fig. 3a-c), but they also showed the decreasing trends well in accordance with global warming. Unlike the melting season, the freezing season (Dec. – Mar.) showed relatively lower ACCs in the marginal and Arctic center regions (Fig. 3g-i). The persistence model did not catch the decreasing trend and*
*showed a negative correlation in the Laptev Sea (Fig. 3g). Further, the ACCs were quite low in the Arctic center region. As mentioned above, the stable and saturated sea ice resulted in lower skill scores in terms of ACC. From visual inspection, the CNN model showed better prediction with a stable skill score than the other models."*

5) The paper needs to be reviewed carefully for proper use of the english language. There are several spelling and grammatical errors.

➔ We corrected the spelling and grammatical errors carefully throughout the entire manuscript.

**Minor comments:**
1) L35-40: "Arctic sea ice has been rapidly declining, which impacts not only the Arctic climate, but also mid-latitudes (Yu et al., 2017)" should say "... but also possibly the mid-latitudes". There is not yet a consensus on this matter.

➔ We revised the statement.

**Lines 36 – 37**: *"Arctic sea ice has been rapidly declining, which impacts not only the Arctic climate but also possibly the mid-latitudes (Yu et al., 2017)."*

2) L40: Usually we say "projecting climate change"; "forecasting" refers to an initial-value problem.

➔ We revised the expression.

**Lines 38 – 39**: *"Therefore, the prediction of long and short-term sea ice change is an important issue in projecting climate change (Yuan et al., 2016)."*

3) L 40-45: Should reference Drobot et al, 2003, 2006; Lindsay et al. 2008; and Wang et al. 2016 for statistical predictions of sea ice concentration.

➔ We have reviewed and added the suggested references regarding the statistical predictions of SIC.

**Lines 43 – 48**: *"The long-range forecasting models of sea ice severity index and concentration*
*(monthly to seasonal) using multiple linear regression were developed by Drobot (2003) and Drobot et al. (2006), respectively. Lindsay et al. (2008) examined the short and long-term sea ice extent prediction using a multiple linear regression model with historical information regarding the ocean and ice data. Wang et al. (2016) developed a vector autoregressive (VAR) model to predict the intraseasonal variability of SIC in the summer season (May – September). The suggested VAR model*
*considering only the historical sea ice data without any atmospheric and oceanic information showed a root mean square error (RMSE) ~ 17% for 30-days' prediction."*

**Added References**:
*"Drobot, S.: Long-range statistical forecasting of ice severity in the Beaufort–Chukchi Sea., Weather*
*Forecast, 18(6), 1161-1176., doi: 10.1175/1520-0434(2003)018<1161:lsfois>2.0.co;2, 2003."*
*"Drobot, S. D., Maslanik, J. A. and Fowler, C.: A long-range forecast of Arctic summer sea-ice minimum extent., Geophys. Res. Lett., 33(10), doi: 10.1029/2006GL026216, 2006."*
*"Lindsay, R. W., Zhang, J., Schweiger, A. J., and Steele, M. A.: Seasonal predictions of ice extent in the Arctic Ocean., J. Geophys. Res.-Oceans., 113(C2)., doi:10.1029/2007JC004259, 2008."*
*"Wang, L., Yuan, X., Ting, M., and Li, C.: Predicting summer Arctic sea ice concentration intraseasonal variability using a vector autoregressive model., J. Climate., 29(4), 1529-1543., doi: 10.1175/JCLI-D-15-0313.1, 2016."*

4) L55-60: "Studies regarding short-term sea ice forecasting have received relatively little attention
(Grumbine, 1998; Preller and Posey, 1989)." Specify that you're referring to machine learning predictions; studies have considered this with classic statistics models (e.g. references on previous comment).

➔ We revised the sentence.

**Lines 61 – 62**: *"However, different from the classic statistical models, the previous studies using deep learning techniques have focused on the long-term prediction of SIC (i.e., over one-year prediction)."*

5) L60: "SIC describes the amount of the sea that is covered by ice". 'The sea' is poorly defined; better to say "SIC describes the fraction of a specified area (typically a grid cell) covered by sea ice".

➔ We revised the definition of SIC as you suggested.

**Lines 64 – 65**: *"SIC describes the fraction of a specified area (typically a grid cell) covered by sea*
*ice."*

6) L185: eqs 1-4; please expand on how these error metrics are computed with respect to space and time. I would think that the spatial averaging would have to be done before any temporal averaging in order for the mask (based on the previous nine years w.r.t the forecast month) to be applied
effectively… Is this correct?

➔ Yes, it is. We revised the sentence about the computation of error metrics.

**Lines 204 – 205**: *"Every error matrix was computed with respect to space and time. The errors were*
*spatially averaged after masking, and then temporally averaged."*

7) L250: "The model did not catch well the decreasing trends of sea ice due to global warming." The model is a linear trend, so that's exactly what it should be doing unless the actual trend is accelerating. I would think It's more likely that the trend forecast is showing 100% SIC in the central Arctic in
years like 2007 and 2012 when sea ice retreated further north than the marginal ice zone predicted by the trend. However, the fact that high-SIC values that are less than 100% do not show this bias in Fig. 3a is highly suspect. Are the authors sure an error hasn't been made either making the figure or in calculating the reference forecast?

➔ Differ to the RF and CNN, the simple yearly extrapolation model showed larger than 100%
of SICs according to their yearly trend. We post-processed them into 100% and the results were shown like in Fig. 3a. There was no mistake in the process.

➔ As we revised the simple prediction model to the anomaly persistence forecast model, the manuscript, as well as Figure 3, were revised either.

8) Fig. 5: Was the impulse noise (i.e. setting to zero) used on the predictors over the training sample and the testing (i.e. forecast) sample?

➔ We adjusted the impulse noise only for the test.

Are the predictions for 2007 and 2012 with impulses on the SIC variables (panels b and c) just that there is no sea ice (all very negative values)?

➔ The model predicted the existence of SICs, but under-estimated. Since the error bar shows only from -30% to 30%, the Figure might result in a confusion. We revised the figure as
below.

[Figure]

Figure 11. The prediction errors (predictions by CNN – NSIDC, %) and RMSE (%) from three prediction results in (a-c) September 2007 (d-f) and 2012: (a and d) original model, (b and e) with noises on SIC variables (sic_1y, sic_1m, ano_1y, and ano_1m), (c and f) with noises on the other variables.

If the impulses are applied over the training data, it's hard to imagine why the remaining variables
wouldn't be capable of "creating ice" in the model, even if it's just a climatology.

➔ The impulses were not applied to the training data. As mentioned in our response to the previous comment, the model predicted the existence of SICs, but under-estimated. We revised Figure 11 to show the effect of impulse noises clearly.

[revised manuscript text omitted]

---

## Author Comment (AC2) · 15 Nov 2019

The authors would like to thank the referees for their precious time and invaluable comments. The corresponding changes and refinements are highlighted in yellow in the revised paper and are also summarized in our responses below. Authors' responses are in blue. Reviewer's comments are in black. When the manuscript in cited, it is shown in italics.

**Response to anonymous referee #2**

**General comments:**
The paper presents a new one-month sea ice concentration prediction model using the Convolutional Neural Network deep learning approach. Output is compared to the results of a Random Forest and a
simple prediction model. Models are applied to the time period from 1988 to 2017 and extreme cases of sea ice concentration decline are analysed in detail.

The subject is appropriate for TC. The title reflects the content of the paper, the abstract provides a complete summary and the paper is generally well structured. The review of existing published work
is good, the number of references is appropriate.

Overall, figures and tables are clear and their captions self-explanatory. However, few figures should by improved according to specific comments below.

Especially the selection of predictors is not convincing and should be justified in more detail.

➔ We conducted a feature selection process in the early stage of the study. Including the eight predictors, four additional predictors were used for feature selection to develop the CNN model:
o ice surface temperature (IST), which affects a heat balance that determines the growth or decay of sea ice (Gabison, 1987; Guemas, 2014);
  o mean sea level pressure (MSL), which is a driving force to make wind variability on the Arctic region as well as sea ice drift (Tsukernik et al., 2009; D̈oscher et al., 2010; Guemas, 2014);
o total cloud cover (TCC), which is a proxy of the amount of solar radiation like a FAL (Kay et al., 2008; Kang et al., 2014);
  o and 10-meter u-wind vector (u-wind), which transfers heat energies across to the Arctic region and affects on growth or decay of sea ice (Arfeuille et al., 2000; Guemas, 2014).
➔ Then we selected predictors using mean decrease accuracy (MDA) based on random forest.
The MDA has widely used feature selection criteria by measuring the accuracy changes by randomly permuting input variables (Strobl et al., 2007; Archer and Kimes, 2008).
➔ Finally, we selected the eight predictors based on the mean MDA from twelve monthly prediction models from 1988 to 2017 using the RF model (Supplementary Table 1 below).

Supplementary Table 1. MDA for input variables used for feature selection in random forest

| ano_1y | t2m | ano_1m | fal | sst | v-wind | sic_1y | sic_1m | tcc | u-wind | msl | ist |
|---|---|---|---|---|---|---|---|---|---|---|---|
| 7.32 | 6.93 | 6.09 | 4.26 | 4.17 | 4.02 | 3.51 | 3.29 | 2.89 | 2.45 | 1.78 | 1.24 |

(references)
Gabison, R. (1987). A thermodynamic model of the formation, growth, and decay of first-year sea ice. Journal of Glaciology, 33(113), 105-119.
Guemas, V., Blanchard-Wrigglesworth, E., Chevallier, M., Day, J. J., Déqué, M., Doblas-Reyes, F. J.,

**Authors' responses (tc-2019-159)**

Fučkar, N. S., Germe, A., Hawkins, E., Keeley, S. and others: A review on Arctic sea-ice predictability and prediction on seasonal to decadal time-scales, Q. J. R. Meteorol. Soc., 142(695), 546–561, doi:10.1002/qj.2401, 2016.

Tsukernik M, Deser C, Alexander M, Tomas R. 2009. Atmospheric forcing of Fram Strait sea ice export: A closer look. Clim. Dyn. 35: 1349–1360, doi: 10.1007/s003-82-009-0647-z.

D¨oscher R, Wyser K, Meier M, Qian M, Redler R. 2010. Quantifying Arctic contributions to climate predictability in a regional coupled ocean–ice–atmosphere model. Clim. Dyn. 34: 1157–1176, doi: 10.1007/s00382-009-0567-y.

Kay, J. E., L'Ecuyer, T., Gettelman, A., Stephens, G. and O'Dell, C.: The contribution of cloud and radiation anomalies to the 2007 Arctic sea ice extent minimum, Geophys. Res. Lett., 35(8), doi:10.1029/2008gl033451, 2008.

Kang, D., Im, J., Lee, M. I., and Quackenbush, L. J.: The MODIS ice surface temperature product as an indicator of sea ice minimum over the Arctic Ocean. Remote Sens. Environ., 152, 99-108., doi.org/10.1016/j.rse.2014.05.012, 2014.

Arfeuille GL, Mysak A, Tremblay LB. 2000. Simulation of the interannual variability in the wind-driven Arctic sea ice cover 1958–1988. Clim. Dyn. 16: 107–121.

Strobl, C., Boulesteix, A. L., Zeileis, A., & Hothorn, T. (2007). Bias in random forest variable importance measures: Illustrations, sources and a solution. BMC bioinformatics, 8(1), 25.

Archer, K. J., & Kimes, R. V. (2008). Empirical characterization of random forest variable importance measures. Computational Statistics & Data Analysis, 52(4), 2249-2260.

➔ We added the justification for the selection of each predictor in the revised manuscript. We also briefly described the feature selection process.

**Lines 86 – 91:** *"In this study, a total of eight predictors were selected and used to predict SIC next month (Table 1) based on the literature and a preliminary statistical analysis of potential predictors through a feature selection process using random forest (Strobl et al., 2007). We selected the eight predictors by comparing the mean decrease accuracy (MDA) changes based on twelve monthly prediction RF models from 1988 to 2017. The MDA has been widely used as feature selection criteria by measuring the accuracy changes by randomly permuting input variables (Archer and Kimes, 2008). It should be noted that fewer predictors than the selected eight ones did not produce better results."*

**Lines 108 – 126:** *"The eight predictors selected in this study though random forest-based feature selection have theoretical backgrounds that are related to the characteristics of SIC. First, SIC itself can affect the SIC in the future because it has a clear inter-annual trend through the melting and freezing seasons (Deser and Teng, 2008; Chi and Kim, 2017). It is a useful characteristic when conducting a time-series analysis, and thus, two SIC time-series climatology predictors (SIC one-year before and SIC one-month before) were used in this study. Although there is no physical explanation of why the interannual variations would contribute to the forecasting skill, it clearly worked well in long-term SIC forecasting through previous studies (Wang et al., 2016; Chi and Kim, 2017). Further, we used two supplementary predictors that indicate the anomalies of SIC one-year before and SIC one-month before, in order to consider anomalous sea ice conditions in the models. The anomaly data could give information about SST anomaly along the sea ice edge in terms of the re-emergence mechanism from the melting to the freezing seasons (Guemas et al., 2014). Second, changes in SST and SIC have a significant relationship with each other, with regards to the heat budget (Rayner et al., 2003; Screen and et al., 2013; Prasad et al., 2018). The re-emergence of sea ice anomalies is also partially explained by the persistence of SST anomalies (Guemas et al., 2014). Air temperature and albedo are related to the amount of solar radiation enabling the prediction of SIC changes. The solar radiation heats the surface of the ocean as well as the sea ice. This causes a rise in the SST while also reducing albedo on the sea ice by melting the surface snow or thinning the sea ice (Screen and Simmonds, 2010; Mahajan et al., 2011). Moreover, the surface snow melting produces melt ponds, wet sea-ice surfaces, and wet snow cover (Kern et al., 2016). Warm winds from lower latitudes toward the Arctic can also reduce sea ice (Kang et al., 2014) and local wind forces affect sea ice*

*motion and formation (Shimada et al., 2006). The wind vector also can cause short or long-range sea ice drifts (Guemas et al., 2014), which may influence SIC variation."*

Regarding the atmospheric predictors, why is FAL and v-wind necessary?

➔ We selected t2m, fal, and v-wind by feature selection process using the MDA analysis mentioned above.
➔ We originally considered TCC, u-wind, and MSL as the atmospheric predictors, but some of them have similar physical backgrounds, which could cause an overfitting problem (i.e., TCC and fal are related with solar radiation, v-wind, u-wind, and MSL are related with wind
parameter because the gradient of surface pressure derives winds).

Why is a simple linear extrapolation model used for a one-month prediction?

➔ We replaced the simple linear extrapolation model to the anomaly persistence forecast model
as requested by referee #1.

**Lines 159 – 160**: *"Finally, an anomaly persistence forecast model was also examined for predicting the monthly Arctic SIC. The anomaly persistence model is a useful reference for forecast skill for time-series data (Wang et al., 2016)."*

**Lines 179 – 182**: *"In the case of the anomaly persistence forecast model, the monthly SIC anomaly of each pixel persisted and the observed trend was calculated for that month ahead. For example, SICs in Jan. 2000 were predicted by summing one-month persisted anomaly and one-month ahead SIC from a linear trend of SICs from Jan. 1988 to Dec. 1999 by each grid."*

**Specific comments:**

1) Figure 1: A larger font should be used in the bottom right part of the figure.

➔  We revised Figure 1 with a larger font.

[Figure]

*Figure 1. Study area and research flow.*

2) Figure 2: Results should be shown for the freeze and the melt season separately.

➔ We revised Figure 2 for the annual mean, melting, and freezing seasons separately.

[Figure]

*Figure 2. The mean absolute SIC anomaly (a) and mean absolute errors between predicted SICs and the actual SICs by the persistence (b), RF (c) and CNN (d) during 2000-2017. As in (a) - (d), but for the melting (Jun. – Sep.) and freezing (Dec. – Mar.) season, (e) - (f) and (i) - (l), respectively.*

3) Figure 8, Figure 9: A larger font should be used.

➔   We revised Figure 8 and 9 with a larger font as suggested.

[Figure]

*Figure 9. Comparison of the prediction results of both models with eight input variables in the Beaufort Sea–Laptev Sea in September 2007. Dotted black circle: the region shows higher prediction errors.*

[Figure]

Figure 10. Comparison of the prediction results of both models with eight input variables in the Barents Sea –Kara Sea in September 2012. Dotted black circle: the region shows higher prediction errors.

**Technical corrections:**
There are numerous spelling and grammatical errors in the text, which should be eliminated.

➔ We corrected the spelling and grammatical errors carefully throughout the entire manuscript.

[revised manuscript text omitted]

---

## Author Response (AR2)

**Response to the comments on "Prediction of monthly Arctic sea ice concentration using satellite and reanalysis data based on convolutional neural networks" by Young Jun Kim et al.**

The authors would like to thank the editor and referees for their precious time and invaluable
comments. The corresponding changes and refinements are highlighted in yellow in the revised paper and are also summarized in our responses below. Authors' responses are in blue. The editor and reviewer's comments are in black. When the manuscript in cited, it is shown in italics.

**Response to editor**

Both referees provided positive feedback for your revised manuscript.
While referee #2 requests only one technical correction, referee #1 mentions a few issues regarding missing literature, restriction to one-month predictions and impact on sea ice extent.

I would like you to improve Figure 1, add the missing literature and to illustrate the impact on sea ice extent. While it would be useful to extend your one-month predictions to seasonal predictions, this might be not feasible within this paper. However, please comment why you restricted your analysis to one-month predictions and whether it is possible in prinicple to extend this method for seasonal predictions.

➔ Thanks for the comment. We revised Figure 1 with larger font size.
➔ We supplemented the explanation of the limitations of the proposed CNN model. We first tested the prediction accuracy of the CNN model by changing prediction lead time from one- to three-months (Supplementary Table 1 below).
➔ Since there are no drastic changes in prediction accuracy among the models, we concluded that the CNN model could be extended to seasonal predictions. However, we need other supplementary variables to build a more accurate long-term prediction model like sea-ice volume and atmospheric circulation (Guemas et al., 2014).

Supplementary Table 1. Average prediction accuracies among three prediction models on the melting season (June – September) during 2000-2017 (mean absolute error, anomaly correlation coefficient, root mean square errors, normalized root mean square errors, and Nash-Sutcliffe efficiency).

|  | One-month | Two-month | Three-month |
|---|---|---|---|
| MAE | 1.96% | 2.71% | 3.00% |
| ACC | 98.09 | 96.51 | 95.71 |
| RMSE | 5.41% | 7.26% | 7.96% |
| nRMSE | 19.09% | 25.60% | 28.26% |
| NSE | 96.14 | 92.99 | 91.34 |

**Lines 470 - 474**: *"The proposed CNN model could be used for the longer prediction (i.e., two- or three-month prediction) in consideration of the persistent effects of input variables such as SST and T2m. Moreover, additional input variables that represent seasonal, or longer-term variabilities of the Arctic environment should be considered in the proposed models. The persistence of sea-ice volume and atmospheric circulation related variables would be suitable for the long-term sea ice forecast*
*(Guemas et al., 2014)."*

[Figure]

Revised Figure 1. Study area and research flow.

**Response to anonymous referee #1**

**Comments:**

1) Lines 40-45. should mention also these statistical models:

Wang, L., Yuan, X., & Li, C. (2019). Subseasonal forecast of Arctic sea ice concentration via statistical approaches. Climate Dynamics, 52(7), 4953–4971.

Kondrashov, D., M. D. Chekroun, and M. Ghil (2018). Data-adaptive harmonic decomposition and prediction of Arctic sea ice extent, Dynamics and Statistics of the Climate System, 3(1).

➔ Thanks for the comment. We have reviewed and added the suggested references regarding the statistical predictions of SIC.

**Lines 50 – 59:** *"A short-term forecast of SIC has been also examined using statistical approaches. Wang et al. (2019) evaluated the sub-seasonal predictability of Arctic SIC using multi-variables of sea ice, the atmosphere, and the ocean based on statistical approaches—the VAR and vector Markov models. The VAR model showed quite good predictability in the short-term with RMSE of 10%, but still resulted in high RMSEs (~20%) for longer than 4 weeks over pan-Arctic during the summer season (from June to August). Meanwhile, the Data-Adaptive Harmonic (DAH) technique, which examines a data-driven feature using cross-correlations, was demonstrated to predict Arctic SIE (Kondrashov et al., 2018). The DAH model showed a promising predictability of SIE in September, resulting in the absolute error of about 0.3 million $km^2$ in 2014-2016."*

2) Authors need to clarify why they have restricted their results to one-month lead time, i.e. for monthly data it is basically one-step prediction.

Can the presented method be used for predicting on longer lead times, i.e. several months ahead, such as for summertime prediction in Sea Ice Outlook (see next)? It is not clear why not, and that begs the question why results are shown for one-month only, thus creating an impression of incomplete study.

➔ We supplemented the explanation of the limitations of the proposed CNN model. We first tested the prediction accuracy of the CNN model by changing prediction lead time from one- to three-months (Supplementary Table 1 below).

➔ Since there are no drastic changes in prediction accuracy among the models, we concluded that the CNN model could be extended to seasonal predictions. However, we need other supplementary variables to build a more accurate long-term prediction model like sea-ice volume and atmospheric circulation (Guemas et al., 2014).

Supplementary Table 2. Average prediction accuracies among three prediction models on the melting season (June – September) during 2000-2017 (mean absolute error, anomaly correlation coefficient, root mean square errors, normalized root mean square errors, and Nash-Sutcliffe efficiency).

|  | One-month | Two-month | Three-month |
|---|---|---|---|
| MAE | 1.96% | 2.71% | 3.00% |
| ACC | 98.09 | 96.51 | 95.71 |
| RMSE | 5.41% | 7.26% | 7.96% |
| nRMSE | 19.09% | 25.60% | 28.26% |
| NSE | 96.14 | 92.99 | 91.34 |

**Lines 470 - 474**: *"The proposed CNN model could be used for the longer prediction (i.e., two- or three-month prediction) in consideration of the persistent effects of input variables such as SST and T2m. Moreover, additional input variables that represent seasonal, or longer-term variabilities of the Arctic environment should be considered in the proposed models. The persistence of sea-ice volume and atmospheric circulation related variables would be suitable for the long-term sea ice forecast*
*(Guemas et al., 2014)."*

3) It would be very helpful to illustrate how prediction of SIC translates into the one for sea ice extent (SIE), and in particular for summertime prediction of September minimum of pan-Arctic SIE which is the main focus of Sea Ice Outlook community effort (Stroeve et al. 2014). Adding and showing
results for one-month prediction (i.e. from August) and observed September pan-Arctic SIE, as well as skill in comparison with RF and baseline prediction model, would be illuminating.

➔ We analyzed the SIE in September 2017 by comparing three models and added Figure 5 to figure out the spatial distributions of sea ice. In addition, we evaluated the proposed
prediction models by comparing other SIO contributions reported in August 2017.

**Lines 233 – 240**: *"The Sea Ice Outlook (SIO) open community has investigated the pan-Arctic sea ice especially in the September SIE since 2008 (Stroeve et al., 2014; Chi and Kim, 2017). They have shared the predicted September SIE from June, July, and August based on a heuristic, statistical,*
*dynamical, and mixed approaches. Chi and Kim (2017) have pointed out the difficulties of sea ice prediction because the prediction errors have increased since 2012. To figure out September minimum SIE which is the main focus of the SIO community (Stroeve et al., 2014), we compared the predicted SIEs based on the three models evaluated in this study, together with the other 37 SIO contributions for the September SIE predictions reported in August 2017. In the present study, the SIE*
*was identified as an area of SIC > 15% (Chi and Kim, 2017)."*

**Lines 325 – 340**: *"The spatial comparison of the predicted September SIEs in 2017 between the reference (NSIDC) and three approaches used in this study is shown in Figure 5. The observed SIE in Sep. 2017 was 4.80 million km2 which was reported by the Sea Ice Prediction Network*
*(http://www.arcus.org/sipn). The SIE in Sep. 13, 2017 was the eighth-lowest in the satellite record since 1981 (NSIDC, 2017). The SIEs predicted by the anomaly persistence, RF and CNN models were 4.37, 4.95, and 4.88 million km2, respectively. While the anomaly persistence model under-estimated the SIE, the other two models slightly over-estimated. The anomaly persistence model considered the decreasing trends of sea ice somewhat excessively. The CNN-based model showed the lowest*
*prediction error compared to the Sea Ice Prediction Network reference data (0.09 million km2). In terms of spatial distributions, the anomaly persistence model showed the excessive retreat of sea ice in the Beaufort and Laptev Sea (Fig. 5a). However, the RF and CNN models showed slightly wide SIE in the Chukchi and Barents Sea (Figs. 5b and c). The over-estimated SIE might be because of the July storm across the central Arctic Ocean through the Barents Sea (West and Blockley, 2017). The*
*accuracy of one-month SIE prediction based on three approaches was compared to the other 37 SIO contributions for Sep. 2017 (Fig. 5d). Since the SIO reports contain only quantitative SIE values, it was not possible to compare their spatial distributions. With regard to the SIE values, the statistical approaches showed quite accurate prediction results based on Arctic sea ice thickness distributions and ice velocity data (UTokyo) and non-parametric statistical model (Slater/Barrett NSIDC). The*
*CNN prediction result showed relatively accurate prediction accuracy."*

[Figure]

Figure 2. The predicted SIEs using the anomaly persistence (a), RF (b), and CNN (c) for Sep. 2017. Distribution of SIO values for Sep. 2017 SIEs reported in Aug. 2017. (d).

**Response to anonymous referee #2**

**A comment:**
1) Accept after revision of figures as suggested.

➔ We revised Figure 1 with a larger font.

[Figure]

Revised Figure 3. Study area and research flow.